# UtilGen: Utility-Centric Generative Data Augmentation with Dual-Level Task Adaptation

**Jiyu Guo**[1]    **Shuo Yang**[1][*]    **Yiming Huang**[1]    **Yancheng Long**[1]    **Xiaobo Xia**[2,3]
**Xiu Su**[4]    **Bo Zhao**[5]    **Zeke Xie**[6]    **Liqiang Nie**[1]

[1]Harbin Institute of Technology, Shenzhen    [2]National University of Singapore
[3]MoE Key Laboratory of Brain-inspired Intelligent Perception and Cognition,
University of Science and Technology of China
[4]Central South University    [5]Shanghai Jiao Tong University
[6]Hong Kong University of Science and Technology (Guangzhou)
{ 220110126@stu.hit.edu.cn, shuoyang@hit.edu.cn }

## Abstract

Data augmentation using generative models has emerged as a powerful paradigm for enhancing performance in computer vision tasks. However, most existing augmentation approaches primarily focus on optimizing intrinsic data attributes – such as fidelity and diversity – to generate visually high-quality synthetic data, while often neglecting task-specific requirements. Yet, it is essential for data generators to account for the needs of downstream tasks, as training data requirements can vary significantly across different tasks and network architectures. To address these limitations, we propose UTILGEN, a novel utility-centric data augmentation framework that adaptively optimizes the data generation process to produce task-specific, high-utility training data via downstream task feedback. Specifically, we first introduce a weight allocation network to evaluate the task-specific utility of each synthetic sample. Guided by these evaluations, UTILGEN iteratively refines the data generation process using a dual-level optimization strategy to maximize the synthetic data utility: (1) model-level optimization tailors the generative model to the downstream task, and (2) instance-level optimization adjusts generation policies – such as prompt embeddings and initial noise – at each generation round. Extensive experiments on eight benchmark datasets of varying complexity and granularity demonstrate that UTILGEN consistently achieves superior performance, with an average accuracy improvement of 3.87% over previous SOTA. Further analysis of data influence and distribution reveals that UTILGEN produces more impactful and task-relevant synthetic data, validating the effectiveness of the paradigm shift from visual characteristics-centric to task utility-centric data augmentation.

## 1 Introduction

Recent advances in generative models, particularly diffusion models [1, 2, 3, 4, 5, 6, 7], have significantly advanced data augmentation by enabling the creation of photorealistic images. Such text-to-image systems are capable of generating diverse and high-fidelity samples, and empirical evidence has shown their potential to enhance downstream model performance [8].

Current generative data augmentation approaches can be categorized into two main paradigms: *fidelity preservation* and *diversity enhancement*. The former employs techniques such as LoRA-based fine-tuning [9] to align synthetic data with real-world distributions [10, 11], while the latter employs

---

[*]Corresponding author.

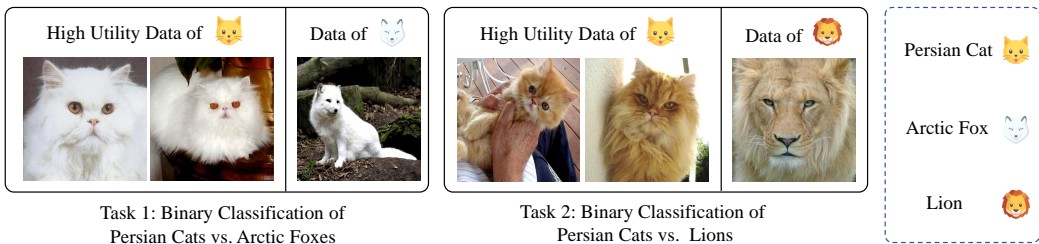

Figure 1: Comparison of high-utility samples within the same category (Persian cats) across two different tasks. White Persian cats (left) are more useful in Task 1, while golden ones (right) are more beneficial in Task 2, highlighting the diverse data requirements in different downstream tasks.

varied prompts or feature perturbations to enhance data diversity [12, 13]. Although effective in generating visually high-quality data, these methods solely optimize the intrinsic data attributes (e.g., fidelity and diversity), often struggling to directly optimize the task-specific utility of synthetic data. In practice, different tasks and model architectures may require distinct training data distributions for optimal performance [14], as exemplified in Figure 1. Despite this, most existing methods lack mechanisms to adapt data generation process based on the needs of specific downstream tasks. This limitation motivates our investigation into utility-centric data augmentation, in which synthetic data is explicitly optimized to enhance task performance rather than merely meet visual standards.

To go beyond the above limitation, an effective mechanism is needed to assess the task-specific utility of synthetic data, thereby providing explicit optimization signals to guide the data augmentation process. However, evaluating utility through full training and testing cycles is computationally prohibitive. Therefore, the core challenges in developing utility-centric data augmentation are: (1) *How to efficiently evaluate the task-specific utility of synthetic data without exhaustive training?* and (2) *How to systematically improve the task-specific utility of synthetic data?*

In this work, we propose UTILGEN, a novel utility-centric data augmentation framework, which can adaptively optimize the data generation process to produce task-specific, high-utility training data based on downstream task feedback. Specifically, we introduce *Task-Oriented Data Valuation*, which quantifies the task-specific utility of synthetic data through a meta-learned weight allocation network [15, 16, 17]. The network is optimized to minimize the classifier's validation loss by adaptively weighting the losses of training samples via estimation of their utility. The trained valuation network serves as an efficient utility predictor, enabling assessment of task-specific utility for newly generated samples without the need for costly retraining cycles. Guided by the utility signals, we employ an integrated dual-level optimization strategy: (1) *Model-Level Generation Capability Optimization* that tailors the data generator to downstream tasks through Direct Preference Optimization (DPO), and (2) *Instance-Level Generation Policy Optimization* that optimizes the generation policies (*i.e.*, prompt embedding and initial noise) to maximize the task-specific utility of synthetic data. Compared to previous advanced data augmentation methods which focus on optimizing intrinsic data characteristics, our proposed method achieves an average accuracy improvement of 3.87% across eight benchmark datasets. To the best of our knowledge, this is the first generative augmentation method where ResNet-50 [18] trained solely on $3\times$ synthetic data surpasses its real-data-trained counterpart on several benchmarks. Before delving into the details, we summarize our contributions as follows:

- Motivated by the observation that training data requirements differ across tasks and network architectures, we introduce a novel paradigm shift in data augmentation. Instead of focusing solely on optimizing intrinsic data attributes, we emphasize enhancing the task-specific utility of synthetic data. This utility-centric approach adaptively optimizes the generation process according to downstream task needs, enabling more targeted and effective data augmentation.

- To efficiently evaluate the task-specific utility of synthetic data, we introduce a meta-learned weight allocation network that measures the utility of synthetic data without requiring costly retraining. These utility signals drive a dual-level optimization framework that enhances both the model generation capability and the generation policies, resulting in high-utility synthetic data tailored to downstream tasks.

- Our method achieves state-of-the-art performance with an average improvement of 3.87% in accuracy across eight benchmarks, while also demonstrating exceptional versatility by delivering

consistent performance gains across diverse architectures (e.g., ResNeXt[19], WideResNet [20], and MobileNet[21]). Through training trajectory analysis, we validate the rationality of task utility measurement based on the weight network. Furthermore, analyses of data influence and distribution reveals that UTILGEN generates data with higher task relevance and stronger positive impact on model performance.

## 2 Related Work

### 2.1 Training Data Valuation

Understanding the role of training data in model performance is crucial for data-efficient learning [22, 23, 24, 25, 26] and optimizing model behavior [27, 28, 29, 30, 31]. To better understand how training data affects model behavior, many studies have aimed to quantitatively assess the influence of individual examples on model performance [32, 33, 34, 35, 36, 37, 38, 39]. Existing approaches for evaluating data value can be classified into two categories: (1) Retraining-based approaches, such as Data Shapley [33, 34, 40, 41] and C-score [35], which quantify data influence through expensive model retraining across different training subsets. For the utility evaluation of large-scale synthetic datasets, these methods become computationally prohibitive due to their inherent complexity. (2) Gradient-based methods [32, 42, 43, 44] that estimate data influence by analyzing gradient interactions between training and test points, either through static snapshot analysis or dynamic trajectory examination. Although these approaches avoid model retraining, they still incur significant computational overhead, particularly when performing complex operations such as Hessian matrix inversion [45]. To evaluate the utility of synthetic data, our method employs a weight allocation network [15, 16, 17] to efficiently assess data utility, avoiding the costly retraining or complex computations required by earlier approaches.

### 2.2 Training Data Augmentation

The availability of high-quality training data has been fundamental to the success of deep learning, enabling models to capture complex patterns, learn meaningful representations, and generate accurate predictions [46, 47, 48, 49, 50]. The methodology of training data augmentation has progressed from traditional techniques to advanced generative approaches [51, 10, 11, 52, 12, 53, 54]. Traditional methods, such as mixup [55, 56], erasing [57, 58], and cropping [59], commonly rely on predefined transformations to augment dataset diversity. However, they are inherently limited to local pixel-level modifications. While Generative Adversarial Networks (GANs) [60] enabled synthetic image generation, they often face challenges in maintaining semantic consistency and distribution alignment [61, 62]. Recent advances in diffusion models, such as Stable Diffusion [4] and GLIDE [63], have demonstrated superior capabilities in generating synthetic data. To enhance diversity, methods such as GIF [12], ALIA [52], and DISEF [13] employ varied prompts or feature perturbations in the latent space. Meanwhile, techniques like RealFake [10], DistDiff [64], and DataDream [11] focus on improving image fidelity by aligning synthetic data distributions with the target domain. Nevertheless, these methods primarily address intra-class distribution alignment without evaluating which types of data better support downstream tasks. In contrast, our approach adaptively tailors the generation process to downstream tasks, producing high-utility data specifically optimized for target applications.

## 3 Method

In this section, the framework of our utility-centric generative data augmentation system is presented, as illustrated in Fig. 2. Specifically, the proposed approach comprises three key components: (1) *Task-Oriented Data Valuation (Sec. 3.1)*, which quantitatively assesses the utility of synthetic data for downstream tasks; (2) *Model-Level Generation Capability Optimization (Sec. 3.2)*, which tailors the generative model to align with the training data preferences of the downstream task via DPO; and (3) *Instance-Level Generation Policy Optimization (Sec. 3.3)*, which adapts generation policy to maximize the task-specific utility of the synthetic data.

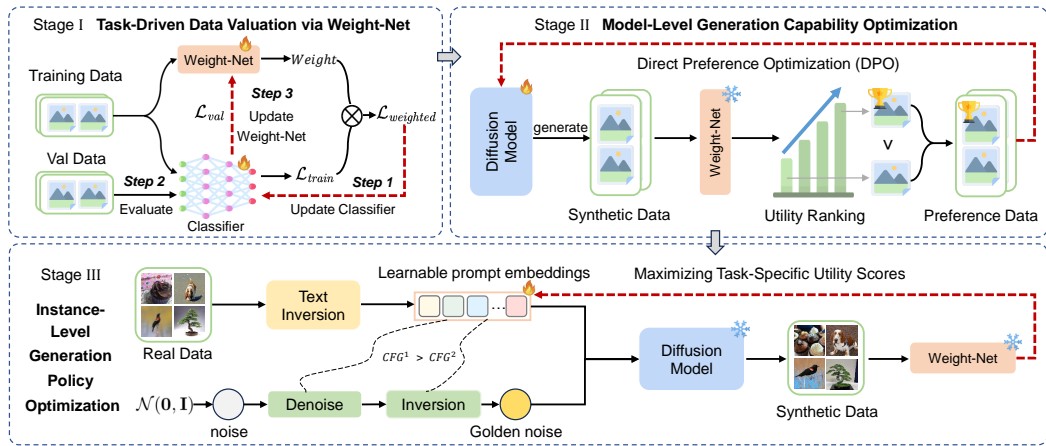

Figure 2: The UTILGEN framework for feedback-driven data augmentation, comprising three key stages: (1) *Task-Oriented Data Valuation* (Sec. 3.1); (2) *Model-Level Generation Capability Optimization* (Sec. 3.2); (3) *Instance-Level Generation Policy Optimization* (Sec. 3.3).

## 3.1 Task-Oriented Data Valuation (TODV)

It is noted that individual training instances exhibit heterogeneous influences on model performance during the training process [32], with certain data samples potentially introducing negative influences. This observation has motivated a line of research to mitigate model overfitting to data samples with negative influences, such as training sample re-weighting strategies implemented through learnable weight networks[15, 16, 17]. Moreover, it is recognized that these learned weights can be implicitly interpreted as indicators of each sample's utility for downstream tasks [15]. Drawing inspiration from this insight, we propose *Task-Oriented Data Valuation*, which employs a weight network trained via meta-learning to quantitatively assess the utility of synthetic data for downstream tasks.

Specially, given a classifier $f_\theta$, the weight $\omega_i$ assigned to each sample $x_i$ is derived through a loss-based process as follows:

$$\omega_i = \mathcal{W}_\phi \left( \mathcal{L}(f(x_i; \theta), y_i) \right), \tag{1}$$

where $\mathcal{W}_\phi$ denotes an MLP network with a single hidden layer. This network is trained to predict normalized weights within the range $[0, 1]$, where higher values reflect samples with greater utility.

To develop such a weight network $\mathcal{W}_\phi$ capable of measuring task-specific data utility, we adopt a bi-level optimization strategy comprising two iterative steps:

**Classifier training:** To enhance the generalization capability of the weight network and mitigate distribution shift in subsequent generation optimization stages, we first perform textual inversion [65] to learn class-specific identifiers $[I_i]$ from a small set of real images per class. Using these learned identifiers, we generate synthetic data $\mathcal{D}_g$ with class-conditioned prompts $c_i$ of the form "a photo of $[I_i]$". The synthetic data is then combined with the real training data to form a merged dataset $\mathcal{D}_{\text{merge}} = (\mathcal{D}_r \cup \mathcal{D}_g)$, where $\mathcal{D}_r$ denotes the real data. The classifier parameters $\theta$ are optimized using a *weighted loss* computed over $\mathcal{D}_{\text{merge}} = \{(x_i, y_i)\}_{i=1}^N$:

$$\theta^*(\phi) = \arg\min_\theta \frac{1}{N} \sum_{i=1}^N \omega_i \mathcal{L}(f(x_i; \theta), y_i), \tag{2}$$

**Weight network training:** Given the classifier's updated parameters, the weight network parameters $\phi$ are trained to minimize the loss on the validation set $\mathcal{D}_v = \{(x_j, y_j)\}_{j=1}^M$:

$$\phi^*(\theta) = \arg\min_\phi \frac{1}{M} \sum_{j=1}^M \mathcal{L} \left( f(x_j; \theta^*(\phi)), y_j \right). \tag{3}$$

This bi-level optimization framework establishes a dynamic feedback loop between data valuation and model training. By quantifying data utility through learned weights, the trained weight network

is subsequently used to guide the optimization of the data generation process. The full procedure for TODV is described in Algorithm 1.

---

**Algorithm 1** Task-Oriented Data Valuation (TODV)

---

**Input**: Training data $\mathcal{D}_r \cup \mathcal{D}_g = \{(x_i, y_i)\}_{i=1}^N$; validation data $\mathcal{D}_v = \{(x_j, y_j)\}_{j=1}^M$
**Required**: Classifier parameters $\theta$; weight network parameters $\phi$; max iteration $T$.

1: Initialize $\theta^{(0)}$, $\phi^{(0)}$ randomly
2: **for** t = 0 to $T - 1$ **do**
3:     Sample batch $\{(x_i, y_i)\}_{i=1}^n$ from $\mathcal{D}_r \cup \mathcal{D}_g$
4:     Sample batch $\{(x_j, y_j)\}_{j=1}^m$ from $\mathcal{D}_v$
5:     Predict weights $\{\omega_i^{(t)}\}_{i=1}^n$ for the batch $\{(x_i, y_i)\}_{i=1}^n$ that reflect their task-specific utility (Eq. 1)
6:     Update $\theta^{(t+1)}$ using the loss weighted by $\{\omega_i^{(t)}\}_{i=1}^n$ (Eq. 2)
7:     Update $\phi^{(t+1)}$ (Eq. 3)
8: **end for**

**Output**: Optimized weight network parameter $\phi^T$

---

## 3.2   Model-Level Generation Capability Optimization (MLCO)

Although standard diffusion models demonstrate remarkable capabilities in generating visually high-quality images, their outputs often fail to meet the specific data requirements of downstream applications. To address this misalignment, we propose an iterative DPO framework that adapts the generative model to downstream task-specific data preferences.

Each optimization cycle begins by prompting the diffusion model with prompts $c_i$ of the form "a photo of $[I_i]$", where class-specific identifiers $[I_i]$ are obtained via textual inversion [65], to generate a synthetic dataset $\mathcal{D}_{\text{syn}} = \{(x_i)\}_{i=1}^M$. The pre-trained weight network $\mathcal{W}_\phi$ subsequently evaluates each sample's utility via the weight score $\omega_i = \mathcal{W}_\phi(\mathcal{L}(f(x_i; \theta), y_i))$, where $\mathcal{L}(f(x_i; \theta), y_i)$ is the loss of the classifier $f_\theta$ on sample $x_i$ with its label $y_i$. Based on these scores, high-utility samples $x_i^w$ and low-utility samples $x_i^l$ are paired to construct a preference dataset:

$$\mathcal{D}_{\text{preference}} = \{(c_i, x_i^w, x_i^l) \mid \mathcal{W}_\phi(\mathcal{L}(f(x_i^w; \theta), y_i^w)) > \mathcal{W}_\phi(\mathcal{L}(f(x_i^l; \theta), y_i^l))\}_{i=1}^N, \tag{4}$$

We then use the preference dataset $\mathcal{D}_{\text{preference}}$ to fine-tune the diffusion model's U-Net $\psi$ using DPO, with the optimization objective formulated according to the Diffusion DPO [66].

$$\mathcal{L}_{\text{DPO}}(\psi) = - \mathbb{E}_{(x_0^w, x_0^l) \sim \mathcal{D}_{\text{preference}}, t \sim \mathcal{U}(0,T), x_t^w \sim q(x_t^w | x_0^w), x_t^l \sim q(x_t^l | x_0^l)} \\ [\log \sigma (-\beta T \omega(\lambda_t) (\Delta\mathcal{L}_w - \Delta\mathcal{L}_l))], \tag{5}$$

$$\Delta\mathcal{L}_w = \|\epsilon^w - \epsilon_\psi(x_t^w, t)\|^2 - \|\epsilon^w - \epsilon_{\text{ref}}(x_t^w, t)\|^2 \\ \Delta\mathcal{L}_l = \|\epsilon^l - \epsilon_\psi(x_t^l, t)\|^2 - \|\epsilon^l - \epsilon_{\text{ref}}(x_t^l, t)\|^2. \tag{6}$$

Here, $\epsilon_\psi$ and $\epsilon_{\text{ref}}$ denote the noise predictions from the trainable and reference U-Nets, respectively. The forward diffusion process $q(x_t | x_0)$ adds noise to $x_0$ at timestep $t$, where $t$ is sampled from $\mathcal{U}(0, T)$. $\beta$ balances preference alignment and KL regularization, while $\sigma(\cdot)$ is the sigmoid activation in the loss. $\lambda_t$ is the signal-to-noise ratio [67] and $\omega(\lambda_t)$ is weighting function [1].

Through iterative DPO fine-tuning, we progressively adapt the diffusion model's generative capability according to the downstream task's preferences for the training data. This process enables better alignment between the model's output distribution and the target application requirements.

## 3.3   Instance-Level Generation Policy Optimization (ILPO)

While MLPO tailors the generative model to the downstream task at a coarse level, ILPO performs fine-grained refinement of the generation policy by jointly optimizing the prompt embeddings and the initial noise. The overall optimization objective is formulated as:

$$(p^*, \epsilon_T^*) = \arg\max_{p, \epsilon_T} \mathbb{E}\left[\mathcal{W}_\phi(\mathcal{L}(f(g(p, \epsilon_T); \theta), y))\right], \tag{7}$$

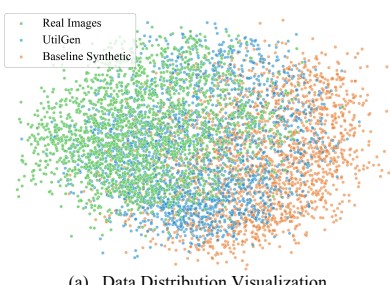 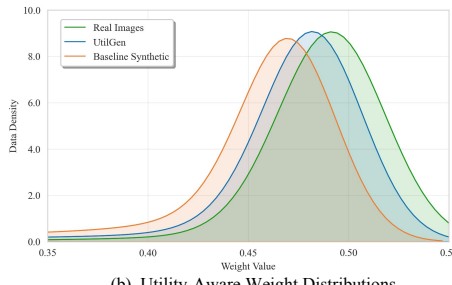

(a) Data Distribution Visualization      (b) Utility-Aware Weight Distributions

Figure 3: (a) Feature space visualization on the Flower dataset [68] shows that our synthetic data achieves closer alignment with the real data distribution compared to vanilla Stable Diffusion. (b) Utility-aware weight distributions for synthetic and real data on the Flower dataset [68], showing sample utility scores for downstream tasks.

---

**Algorithm 2** Instance-Level Generation Policy Optimization (ILPO)

---

**Input**: $K$ classes: $\{c_1, ..., c_K\}$; number of images to generate for each class: $\{N_1, \ldots, N_K\}$, where $N_k$ is the number of samples to generate for class $c_k$

**Required**: Diffusion model $g(\cdot, \cdot)$ fine-tuned by MLCO; Textual Inversion technology $TI(\cdot, \cdot)$ using few-shot real images per class $\mathcal{D}_k = \{x_1^k, ..., x_N^k\}$ for each $c_k$;

1: **for** each class $k \in \{1, ..., K\}$ **do**
2:      // **Prompt Embedding Optimization**
3:      Initialize $p_k \leftarrow TI(c_k, \mathcal{D}_k)$
4:      $p_k^* \leftarrow p_k$ (Eq. 8)                      ▷ Optimize to maximize data utility
5:      // **Noise Optimization**
6:      **for** $i = 1$ to $N_k$ **do**
7:          Sample noise $\epsilon_T \sim \mathcal{N}(0, I)$
8:          $\epsilon_T' \leftarrow \epsilon_T$ by (Eq. 9)      ▷ Inject semantic information representing high-utility data
9:          $x_i' \leftarrow g(p_k^*, \epsilon_T')$                        ▷ Generate high-utility data
10:         $\mathcal{D}_{\text{synth}} \leftarrow \mathcal{D}_{\text{synth}} \cup \{x_i'\}$
11:      **end for**
12: **end for**

**Output**: High-utility synthetic dataset $\mathcal{D}_{\text{synth}}$

---

where $p$ denotes the prompt embedding, $\epsilon_T$ represents the initial noise vector, $g(\cdot)$ is the diffusion model, $f(\cdot; \theta)$ is the downstream classifier, $y$ is the ground-truth label, and $\mathcal{L}(\cdot)$ is the classification loss. The optimization process consists of two synergistic components:

**Prompt embedding optimization:** Building upon the class-specific identifiers $[I_i]$ learned through textual inversion [65], we optimize the prompt embeddings $p_i = E(\text{"a photo of } [I_i]\text{"})$ using gradient-based optimization, where $E(\cdot)$ denotes the text encoder. This process aims to maximize the utility score predicted by the pre-trained weight network. To preserve semantic alignment during optimization, a CLIP-based regularization term $L_{\text{CLIP}} = -\cos(E(x_i), e_i)$ is applied, where $x_i$ is the generated image, $E(\cdot)$ is the CLIP image encoder and $e_i = \text{avg}(E(\{x_j\})$ is the mean embedding of a set of target-class real images $\{x_j\}$. The complete prompt optimization objective is:

$$p^* = \arg\max_p \left[ \mathcal{W}_\phi(\mathcal{L}(f(g(p, \epsilon_T); \theta), y)) - \lambda L_{\text{CLIP}} \right], \tag{8}$$

where $\lambda$ controls the regularization strength. This joint optimization enhances sample utility while preserving semantic coherence with the original class concept.

**Noise optimization:** The quality of synthetic images is influenced by both the text prompt and the random Gaussian noise. Yet, since each image generation requires independently sampled noise, directly optimizing noise vectors via gradient ascent to improve utility is computationally prohibitive. Recent studies [69, 70, 71] show that the discrepancy between denoising and inversion Classifier-Free Guidance (CFG) scales can be leveraged to implicitly inject prompt semantic information into the initial noise. Leveraging this observation, we adapt the methodology to optimize the initial noise, enabling it to incorporate semantic information of high-utility data. Formally, the noise optimization

Table 1: Classification performance across eight datasets using ResNet-50 [18] as the backbone classifier. UTILGEN and DataDream [11] generate synthetic data guided by **16-shot real images** per class, while other methods use the full real dataset as guidance for generation. The top block shows results using synthetic data only, while the bottom includes joint training (synthetic + full real dataset). Each method produces synthetic data at $5\times$ the scale of the original real dataset.

| Method | Real | Syn | IN-1k-S | IN-100-S | Cal101 | DTD | CUB | PETs | Food-S | Flowers | Avg |
|---|---|---|---|---|---|---|---|---|---|---|---|
| | | | | | Training on Synthetic Data Only | | | | | | |
| SD v2.1 [4] | | ✓ | 24.35 | 27.96 | 14.74 | 7.92 | 23.43 | 26.05 | 24.02 | 31.41 | 22.49 |
| GIF [12] | | ✓ | 28.95 | 31.94 | 20.39 | 13.19 | 27.54 | 29.73 | 25.94 | 56.12 | 29.23 |
| GAP [72] | | ✓ | 25.84 | 30.94 | 18.70 | 11.01 | 29.49 | 27.46 | 25.31 | 53.66 | 27.80 |
| DataDream [11] | | ✓ | 30.35 | 35.48 | 23.61 | 13.24 | 35.38 | 34.77 | 28.41 | 65.15 | 33.30 |
| **UtilGen** | | ✓ | **33.72** | **40.94** | **29.31** | **13.52** | **43.32** | **37.25** | **31.87** | **67.43** | **37.17** |
| △ *over previous SOTA* | | | **+3.37** | **+5.46** | **+5.70** | **+0.28** | **+7.94** | **+2.48** | **+3.46** | **+2.28** | **+3.87** |
| | | | | | Joint Training with Real Data | | | | | | |
| Real Dataset | ✓ | | 36.34 | 38.58 | 43.55 | 16.32 | 21.03 | 28.78 | 19.88 | 73.34 | 34.73 |
| SD v2.1 [4] | ✓ | ✓ | 43.26 | 49.86 | 59.35 | 28.82 | 43.52 | 52.35 | 40.32 | 79.58 | 49.63 |
| GIF [12] | ✓ | ✓ | 49.85 | 54.12 | 67.60 | 33.45 | 43.80 | 58.21 | 41.39 | 84.47 | 54.11 |
| GAP [72] | ✓ | ✓ | 46.58 | 53.14 | 66.97 | 32.87 | 47.46 | 57.86 | 43.77 | 84.84 | 54.19 |
| DataDream [11] | ✓ | ✓ | 52.16 | 57.68 | 73.38 | 34.84 | 53.43 | 60.83 | 47.44 | 89.60 | 58.67 |
| **UtilGen** | ✓ | ✓ | **54.56** | **61.54** | **75.62** | **36.06** | **57.53** | **64.64** | **52.72** | **93.62** | **62.04** |
| △ *over real dataset* | | | **+18.22** | **+22.96** | **+32.07** | **+19.74** | **+36.50** | **+35.86** | **+32.84** | **+20.28** | **+27.31** |
| △ *over previous SOTA* | | | **+2.40** | **+3.86** | **+2.24** | **+1.22** | **+4.10** | **+3.81** | **+5.28** | **+4.02** | **+3.37** |

process is defined as:

$$\epsilon'_t = \text{DDIM-Inversion}_{\omega_w}(\text{DDIM}_{\omega_l}(\epsilon_t, p^*)). \tag{9}$$

where $\omega_l$ and $\omega_w$ are the CFG scales for the denoising process $\text{DDIM}(\cdot)$ and the inversion process $\text{DDIM-Inversion}(\cdot)$, respectively. The condition $\omega_l > \omega_w$ enables the implicit injection of semantic information into the initial noise. The entire process of ILPO is outlined in Algorithm 2.

The synergistic integration of MLCO (Sec.3.2) and ILPO (Sec.3.3) enables UTILGEN to synthesize data that closely aligns with real data feature distributions, as visualized in Fig.3(a). This approach simultaneously achieves higher utility scores (Fig.3(b)), demonstrating enhanced task relevance and superior data utility in downstream applications.

## 4 Experiments

### 4.1 Experimental Setup

**Benchmarks.** We evaluate the effectiveness of UTILGEN across eight datasets spanning three classification tasks: coarse-grained classification (ImageNet-1k-Subset [73], ImageNet-100-Subset [73], and Caltech 101 [74]), fine-grained classification (Oxford Pets [75], Food-S [76], Flowers 102 [68], and CUB-200-2011 [77]), and texture classification (DTD [78]). Specifically, ImageNet-1k-Subset and Food-S are subsets of ImageNet-1K and Food101 [76], respectively, each containing 100 randomly selected images per class. ImageNet-100-Subset is constructed by randomly sampling 100 animal-related classes from the original ImageNet-1K [73], with 100 randomly selected images per class. Further benchmark details are provided in Appendix F.

**Baselines.** To compare our utility-centric approach with existing methods that focus on optimizing intrinsic data characteristics, we select GIF [12] and DataDream [11] as representative baselines. Specifically, GIF [12] enhances diversity by applying feature perturbations, while DataDream [11] improves fidelity through domain alignment using LoRA fine-tuning of the diffusion model. Additionally, we include GAP [72], which uses feedback from the downstream model to generate adversarial prompts that maximize the model's loss on the generated images. This enables a direct comparison between its loss-based feedback strategy and the utility-based feedback mechanism employed by UTILGEN. For fair comparison, we adopt Stable Diffusion v2.1 [4] (SD v2.1) as the backbone generator across all baseline methods. The implementation details of UTILGEN are in Appendix B.

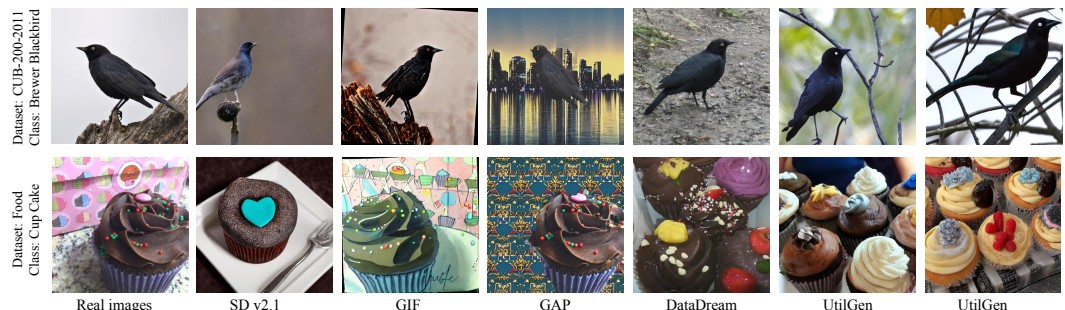

Figure 4: Comparison of synthetic images generated by SD v2.1, GIF [12], GAP [72], DataDream [11], and UTILGEN.

Table 2: Performance comparison under different synthesis budgets on ImageNet-100. UTILGEN scales more effectively with increasing synthetic data ratios.

| Budget | SD v2.1 | GIF [12] | GAP [72] | DataDream [11] | UTILGEN |
|---|---|---|---|---|---|
| 1× | 12.18 | 13.44 | 15.02 | 17.12 | **18.04** |
| 3× | 20.20 | 21.90 | 21.16 | 25.26 | **28.52** |
| 5× | 27.96 | 31.94 | 30.94 | 35.48 | **40.94** |

Table 3: Comparison of intra-class diversity of synthetic data on ImageNet-100. Higher values indicate greater diversity.

| Method | Mean Intra-Class Diversity ↑ |
|---|---|
| Stable Diffusion v2.1 | 0.5815 |
| DataDream [11] | 0.5238 |
| UTILGEN | **0.6054** |

## 4.2 Evaluation Results

**Results on solely synthetic data.** As shown in Table 1 (top), UTILGEN achieves the highest average accuracy of 37.17% when trained solely on synthetic data, outperforming the previous best method DataDream (33.30%) by a notable margin of +3.87%. It demonstrates strong performance across both coarse-grained (e.g., 40.94% on IN-100-S [73]) and fine-grained tasks (e.g., 67.43% on Flowers [68]), indicating excellent generalization despite training solely on synthetic data.

**Results on real + synthetic Data.** In the joint training setting (bottom of Table 1), UTILGEN maintains its lead with an average accuracy of 62.04%, surpassing DataDream (58.67%) by +3.54%. It achieves particularly strong gains on fine-grained datasets (e.g., 93.62% on Flowers [68]), while also delivering consistent improvements on coarse-grained tasks. These results suggest that UTILGEN can effectively complement real data across different task granularities, providing high-utility synthetic samples that enhance model performance and robustness.

**Synthetic data scaling effects.** UTILGEN exhibits strong scalability and data augmentation efficiency. As shown in Figure 5, scaling synthetic data from 1× to 5× the original set consistently improves ResNet-50 [18] performance across benchmarks under both synthetic-only and joint training settings. When trained with 3× synthetic data alone, models outperform their real-data-trained counterparts on three datasets; at 5× scaling, this advantage extends to four datasets. To further examine how performance scales with the synthesis budget, we compare UTILGEN with four representative baselines under budgets. As shown in Table 2, UTILGEN consistently achieves the best results across all budgets, and the performance gap widens as the synthesis ratio increases, demonstrating superior scalability under large-scale generation. Additional details on augmentation efficiency and computational cost are provided in Appendix D.

**Diversity analysis of synthetic data.** We evaluate the diversity of synthetic samples by computing the mean intra-class cosine distance of CLIP (ViT-L/14) features on ImageNet-100. Compared methods include vanilla Stable Diffusion v2.1, DataDream [11], and UTILGEN. Higher values indicate greater intra-class diversity, which typically benefits generalization. As shown in Table 3, UTILGEN achieves the highest intra-class diversity, indicating that its utility-guided optimization preserves sample variety and avoids mode collapse.

**Reusability of synthetic data across tasks.** We further analyze the reusability of UTILGEN-generated data across different downstream models on the ImageNet-100 dataset. Even when the weight network is trained using ResNet-50 , the resulting synthetic data generalizes effectively to other models such as WideResNet and CLIP. As shown in Table 4, the performance gains remain

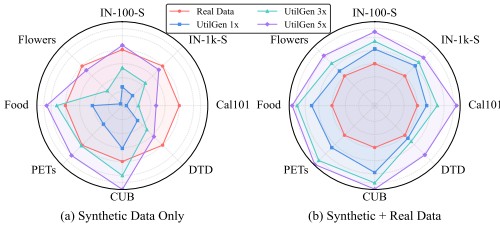
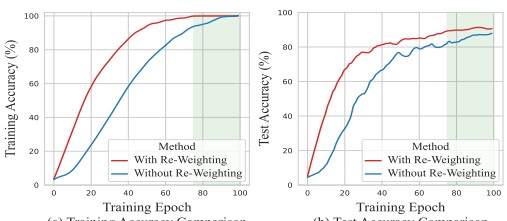

Figure 5: Synthetic Data Scaling Effects across different training data regimes. (a) Models trained exclusively on synthetic data (Synthetic Data Only). (b) Models trained on combined synthetic and real data (Synthetic + Real Data).

Figure 6: Training and test accuracy comparison with and without the weight network using ResNet-50 [18] on the dataset (original Flowers + SD v2.1 augmented data with $1\times$ expansion). (a) Training convergence; (b) Test accuracy.

Table 4: Reusability of synthetic data across downstream models on ImageNet-100.

| Classifier used to train weight network | Downstream model using synthetic data | Method | Acc. (%) |
|---|---|---|---|
| – | WideResNet | DataDream [11] | 31.76 |
| ResNet-50 | WideResNet | UTILGEN | **36.40** |
| – | CLIP | DataDream [11] | 71.42 |
| ResNet-50 | CLIP | UTILGEN | **72.14** |

Table 5: Generalization performance across diverse network architectures on ImageNet-100.

| Method | ResNeXt-50 | WideResNet-50 | MobileNetV2 |
|---|---|---|---|
| GIF [12] | 27.54 | 27.84 | 31.24 |
| GAP [72] | 27.66 | 27.76 | 32.72 |
| DataDream [11] | 31.24 | 31.76 | 35.48 |
| **UTILGEN** | **37.62** | **37.82** | **40.59** |

consistent across architectures, indicating that high-utility samples identified by UTILGEN are not tied to a specific model and can be reused for diverse learning objectives.

**Generalization across different architectures.** To assess the versatility of UTILGEN, we evaluate its performance on three architectures: ResNeXt-50 [19], WideResNet-50 [20], and MobileNetV2 [21]. As shown in Table 5, UTILGEN consistently achieves the highest accuracy, with 500 images generated per class, confirming the effectiveness of our approach across various network architectures.

**Cost-benefit analysis compared to collecting more labeled data.** Compared to manual data collection and annotation, UTILGEN offers a highly cost-effective solution for dataset expansion. According to the Masterpiece Group[2], manually annotating 10,000 images (e.g., 100 images per class across 100 classes) typically takes about two weeks and costs approximately $800. In contrast, generating the same amount of data using UTILGEN requires only about 0.94 hours and $20 on 8 V100 GPUs rented from Google Cloud[3]. On the ImageNet-100-Subset dataset, using $5\times$ synthetic data produced by UTILGEN even surpasses the real-data baseline in accuracy, while requiring only $\sim$4.7 hours and about $100 in compute cost, as shown in Table 6.

**Ablation study.** Table 7 presents an ablation study on the IN-100-S dataset using ResNet-50 trained solely on synthetic data (500 aasssper class). We analyze the effects of MLCO and ILPO (including prompt embedding and initial noise optimization). Each component brings improvements over the baseline, and combining all three achieves the best performance, surpassing the baseline by +12.98%, demonstrating the complementary strengths of model-level and instance-level optimizations.

### 4.3 Mechanism Analysis

**Effect of sample re-weighting on classifier training.** To assess the effect of the weight allocation network on classifier training, we compare the training trajectories of models with and without dynamic weighting. As shown in Figure 6(a), the weighted model converges faster and achieves higher training accuracy in earlier epochs, suggesting it effectively assigns higher weights to high-utility samples during learning. Figure 6(b) presents the test accuracy curves, where accuracy rises more rapidly and achieves a final improvement of 2.94% over the baseline. This gain is attributed to the network's ability to down-weight low-utility or noisy samples, thereby mitigating their negative

---

[2]https://mpg-myanmar.com/annotation/

[3]https://cloud.google.com/compute/gpus-pricing

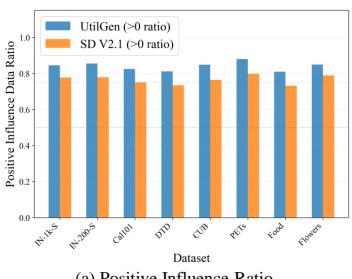 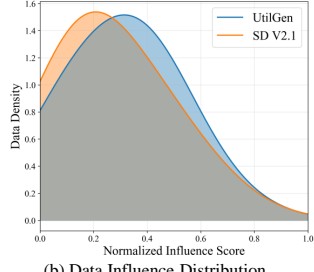 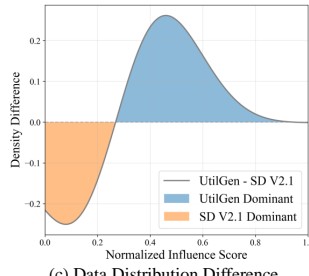

(a) Positive Influence Ratio    (b) Data Influence Distribution    (c) Data Distribution Difference

Figure 7: Data influence comparison between UTILGEN and SD v2.1, computed using influence functions. (a) Proportion of samples with positive influence scores (influence $> 0$) across eight datasets. (b) Influence score distributions showing that UTILGEN generates a higher density of samples with stronger influence. (c) Data density difference plot highlighting that UTILGEN dominates in high-influence regions, while SD v2.1 contributes more to low-influence areas.

Table 6: Comparison of the manual annotation paradigm and our synthetic data paradigm in terms of cost, time, and performance.

| Method | Images | Time Required | Cost | Acc. (%) |
|---|---|---|---|---|
| Manual Annotation | 10,000 | ~2 weeks | $800 | 38.58 |
| UTILGEN (1×) | 10,000 | ~0.94 h | $20 | 18.04 |
| UTILGEN (5×) | 50,000 | ~4.70 h | $100 | **40.94** |

Table 7: Ablation study on ImageNet-100.

| MLCO | Prompt Optimization | Noise Optimization | Acc. (%) |
|---|---|---|---|
| | | | 27.96 |
| | | ✓ | 28.68 |
| ✓ | | | 36.42 |
| | ✓ | | 37.96 |
| ✓ | | ✓ | 32.08 |
| | ✓ | ✓ | 39.12 |
| ✓ | ✓ | | 39.73 |
| ✓ | ✓ | ✓ | **40.94** |

impact. These results confirm that the pre-trained weight allocation network can reliably measure sample utility and serve as a signal to guide synthetic data generation.

**Data influence analysis.** We measure data influence by jointly training models on synthetic datasets generated by UTILGEN and the baseline SD v2.1, applying Influence Function [32]. Figure 7(a) reveals that UTILGEN consistently produces a higher proportion of positively influential samples (influence > 0) across all benchmarks. Furthermore, the influence score distribution shown in Figure 7(b) shifts noticeably to the right, indicating that UTILGEN generates more samples with stronger positive influence. Complementing this, Figure 7(c) highlights that UTILGEN achieves substantially higher data density within high-influence regions, while simultaneously reducing sample concentration in low-utility areas. Together, these results validate the effectiveness of our method in synthesizing impactful data that better supports downstream model optimization.

## 5   Conclusion

In this study, we propose UTILGEN, a utility-centric data augmentation framework that shifts the focus from optimizing intrinsic visual properties to enhancing task-specific utility. By incorporating downstream model feedback, UTILGEN adaptively adjusts the data generation process to produce high-utility data tailored for specific downstream tasks, thereby establishing a feedback loop between data generation and model training. Experiments across eight benchmarks demonstrate consistent performance gains, and in certain cases surpassing the performance of models trained solely on real data. These results highlight the superiority of the utility-centric approach over prior methods focusing primarily on intrinsic visual quality. These findings underscore the potential of utility-centric generation and suggest that integrating task-specific utility alongside traditional visual quality considerations offers a more effective paradigm for future data augmentation research.

## Acknowledgment

The authors would like to acknowledge the support from the following funding sources. Shuo Yang is supported by the National Natural Science Foundation of China Young Scientists Fund

(No. 62506096). Xiaobo Xia is partially supported by the MoE Key Laboratory of Brain-inspired Intelligent Perception and Cognition at the University of Science and Technology of China (Grant No. 2421002). Bo Zhao is supported by the National Natural Science Foundation of China (Grant No. 62306046).

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

# A  Additional Visualizations of Synthetic Data

In this section, we present further visualizations of synthetic data generated by the UTILGEN. As illustrated in Figure 8, we compare synthetic samples produced by different augmentation strategies, including GIF (prioritizing diversity in visual features) and DataDream (emphasizing fidelity in visual features). Figure 8 presents representative samples from UTILGEN, demonstrating semantically consistent, visually realistic generations that preserve class-discriminative features while maintaining diversity.

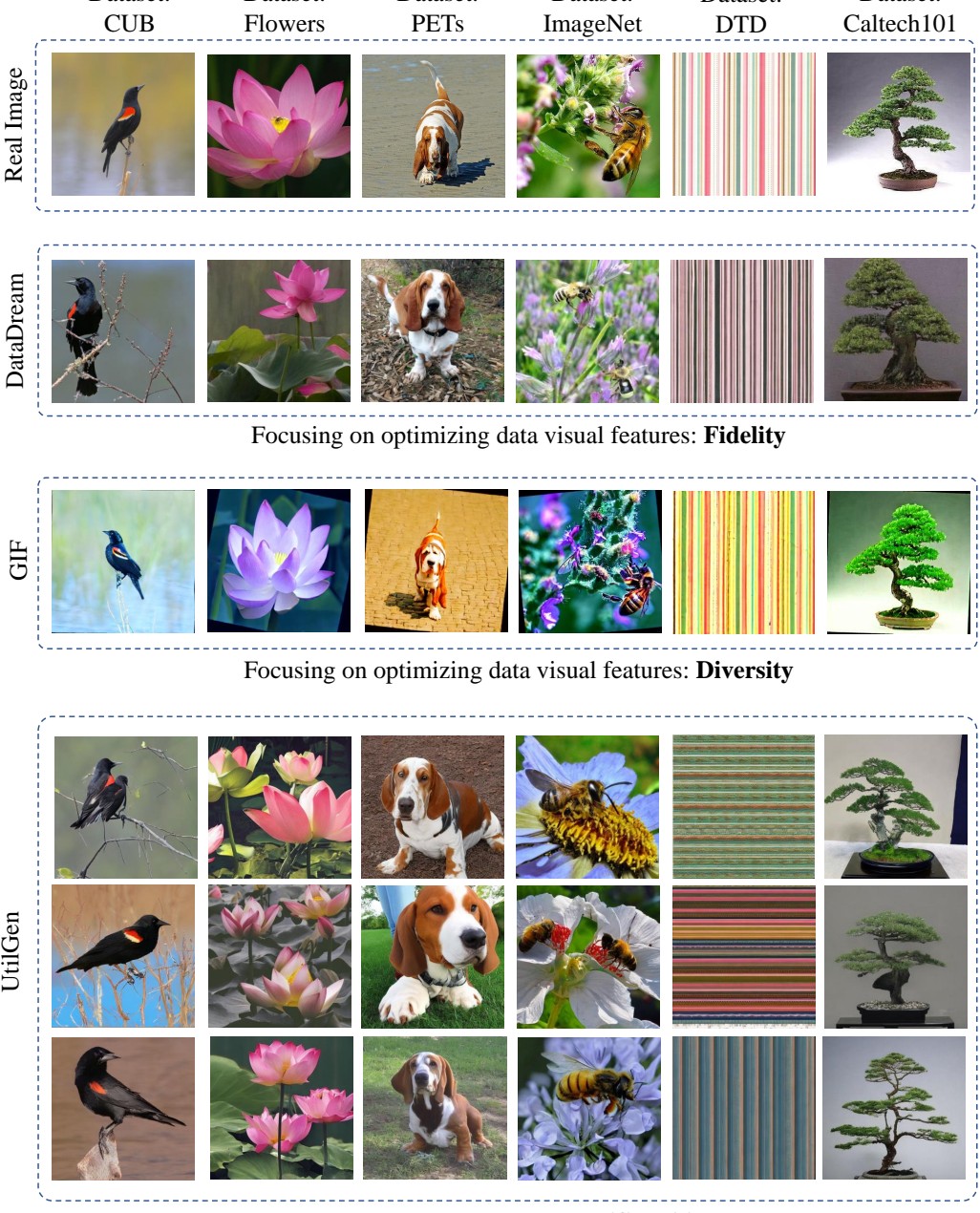

Figure 8: Comparison of synthetic samples generated by GIF, DataDream, and UTILGEN.

# B More Implementation Details

## B.1 Implementation Details of Weight Network

The weight network $\mathcal{W}_\phi$ is designed as a lightweight MLP with a single hidden layer. It takes the per-sample classification loss $\ell_i = \mathcal{L}(f(x_i; \theta), y_i)$ as input, where $f(\cdot; \theta)$ is the downstream classifier, and predicts normalized sample weights $\omega_i \in [0, 1]$ via a sigmoid activation. These weights are used to re-weight the corresponding per-sample losses during classifier training, thereby prioritizing examples deemed to have higher utility by the meta-learned network. The network architecture is defined as:

$$\mathcal{W}_\phi(\ell_i) = \sigma(W_2 \, \mathrm{ReLU}(W_1 \ell_i + b_1) + b_2) \tag{10}$$

where $\ell_i \in \mathbb{R}$ is the scalar loss value for the $i$-th sample, $W_1 \in \mathbb{R}^{1 \times 100}$ and $W_2 \in \mathbb{R}^{100 \times 1}$ are weight matrices, and $b_1, b_2$ are bias terms. The ReLU activation introduces non-linearity, and the final sigmoid ensures the output lies in $[0, 1]$. Albeit simple, this network is a universal approximator for continuous functions and can fit a wide range of weighting functions. The training settings for both the classifier and the weight network, including optimizers, learning rates, and batch sizes, are summarized in Table 8.

Table 8: Training settings for the classifier and weight network.

| Component | Optimizer | Learning Rate | Batch Size |
|---|---|---|---|
| Classifier | SGD (momentum=0.9) | 0.01 | 128 |
| Weight Network | Adam | $10^{-3}$ | 128 |

## B.2 Implementation Details of MLCO

---

**Algorithm 3** Model-Level Generation Capability Optimization (MLCO)

---

**Input**: Initial diffusion model $g_\psi$; class prompts $\{c_1, ..., c_K\}$.
**Required**: Batch size $B$; DPO learning rate $\eta$; reference model $g_{\mathrm{ref}}$; weight network $\mathcal{W}_\phi$; downstream classifier $f_\theta$; loss function $\mathcal{L}$; max iteration $I$; selection ratio $\rho$..

1: **for** iteration = 1 to $I$ **do**
2:     // Generation & Evaluation
3:     **for** each class $k \in \{1, ..., K\}$ **do**
4:         **for** $i = 1$ to $B$ **do**
5:             Sample $\epsilon_T \sim \mathcal{N}(0, I)$
6:             $x_i^k \leftarrow g_\psi(c_k, \epsilon_T)$                           ▷ Generate with current model
7:             $\omega_i^k \leftarrow \mathcal{W}_\phi(\mathcal{L}(f(x_i^k; \theta), y_k))$        ▷ Evaluate utility using classification loss as input
8:         **end for**
9:     **end for**
10:     // Preference Construction
11:     **for** each class $k$ **do**
12:         Sort $\{x_i^k\}$ by $\omega_i^k$ descending
13:         Select top $\rho$ proportion of samples as $\mathcal{D}_k^w$, bottom $\rho$ proportion of samples as $\mathcal{D}_k^l$
14:         $\mathcal{D}_{\mathrm{pref}} \leftarrow \mathcal{D}_{\mathrm{pref}} \cup \{(c_k, x^w, x^l) | x^w \in \mathcal{D}_k^w, x^l \in \mathcal{D}_k^l\}$
15:     **end for**
16:     // Model Optimization
17:     **for** each $(c_k, x^w, x^l)$ in $\mathcal{D}_{\mathrm{pref}}$ **do**
18:         $\psi \leftarrow \psi - \eta \nabla_\psi \mathcal{L}_{\mathrm{DPO}}(g_\psi, g_{\mathrm{ref}}, c_k, x^w, x^l)$        ▷ Update via Eq. 5
19:     **end for**
20: **end for**

**Output**: Optimized diffusion model $g_\psi^*$

---

The proposed Model-Level Generation Capability Optimization (MLCO) framework iteratively improves the diffusion model's generative ability based on utility-guided preferences. In each iteration, the process proceeds in three stages, and he full process is illustrated in Algorithm 3.:

- **Generation & Evaluation:** The current diffusion model $g_\psi$ generates synthetic images for each class prompt. Each generated sample is then evaluated by the trained weight network $\mathcal{W}_\phi$ to obtain utility scores $\omega$.
- **Preference Construction:** Samples within each class are ranked by their utility scores. The top $\rho$ and bottom $\rho$ proportions are selected to form preference pairs $\mathcal{D}_{\text{pref}}$.
- **Model Optimization:** Using Direct Preference Optimization (DPO), the model is updated to customize its generation capability based on utility-guided preferences over training data.

Our implementation of DPO is adapted from Diffusion-DPO [66], which enables preference-based training of diffusion models for guiding generative outputs with utility-informed preferences. The loss function Eq. 5 used in Line 19 of Algorithm 3 follows this framework. For the detailed mathematical derivation of Eq. 5, please refer to the original Diffusion-DPO paper [66]. The key hyperparameters used in our DPO training process are summarized in Table 9.

Table 9: Hyperparameters used in DPO training

| Batch Size | Max Steps Per Class | Learning Rate | Gradient Accumulation Steps | Beta DPO |
|---|---|---|---|---|
| 1 | 400 | $1 \times 10^{-8}$ | 1 | 5000 |

## B.3 Implementation Details of ILPO

The prompt-noise optimization process consists of two primary components: the optimization of prompt embeddings and the initial noise used during generation. The goal is to maximize the utility of each synthetic sample while ensuring semantic alignment with the target domain.

**Prompt embedding optimization:** The optimization process begins with textual inversion [65] to establish class-specific prompt embeddings that align with target concepts. Specifically, we use DeepSeek-R1-Distill-Qwen-1.5B [79] to select an initializer token for textual inversion [65] for each class. After obtaining the class-specific prompt embeddings aligned with the target labels, these embeddings are refined through gradient-based optimization to maximize the utility score of the generated samples. The learning rate for prompt optimization is set to 0.001, and the optimization process runs for 400 epochs to ensure convergence and meaningful results.

Table 10: Hyperparameters used in textual inversion [65]

| Batch Size | Learning Rate | Training Steps | Instance Images per Class |
|---|---|---|---|
| 1 | $1 \times 10^{-4}$ | 400 | 16 |

**Noise optimization:** For noise optimization, we leverage the discrepancy between denoising and inversion Classifier-Free Guidance (CFG) scales, which allows us to implicitly inject prompt semantic information into the noise vector. The denoising guidance scale is set to 5.5, while the inversion guidance scale is set to 0.

The specific hyperparameters used for this optimization process are summarized in Table 11.

Table 11: Hyperparameters used in the prompt-noise optimization process.

| Prompt Learning Rate | Prompt Learning Epochs | Guidance Strength (Denoise) | Guidance Strength (Inversion) |
|---|---|---|---|
| 0.001 | 400 | 5.5 | 0 |

## B.4 Implementation Details of Image Generation

The image generation process is guided by the hyperparameters listed in Table 12. Instead of using fixed prompts and random noise, both the class-specific text prompts and the initial noise vectors

are optimized via our proposed ILPO strategy to better align with high-utility regions in the data distribution. The Stable Diffusion v2.1 model, fine-tuned with MLCO, is employed for the generation process, using a sampling method with 50 steps, the DDIM scheduler, and a guidance scale of 2.0. The images are generated at a resolution of $512 \times 512$ pixels.

Table 12: Hyperparameters for Training Data Synthesis

| Base Model | Sampling Steps | Scheduler | Guidance Scale | Image Size |
|---|---|---|---|---|
| Stable Diffusion v2.1 (MLCO-fine-tuned) | 50 | DDIM | 2.0 | 512×512 |

### B.5 Implementation details of model training

The downstream classifiers are trained on four standard architectures: ResNet-50[18], ResNeXt-50 [19], WideResNet-50 [20], and MobileNetV2 [21]. All models are trained with identical hyperparameters. The training configuration uses SGD optimizer with momentum 0.9 and weight decay 5e-4. The learning rate starts at 0.01 with cosine decay schedule over 100 epochs. A fixed batch size of 256 is used for all experiments, with standard data augmentation including random horizontal flips and crops. Each experiment is repeated three times with different random seeds to ensure reliability.

Table 13: Hyperparameters for Downstream Classifier Training

| Optimizer | Weight decay | Initial LR | Epochs | Batch size |
|---|---|---|---|---|
| SGD (momentum=0.9) | 5e-4 | 0.01 | 100 | 256 |

## C Limitations

While demonstrating strong performance in enhancing synthetic data utility for downstream tasks, UTILGEN presents two noteworthy considerations: (1) The dual-level optimization framework incurs modest computational overhead compared to conventional augmentation methods; (2) Although effectively improving the utility of synthetic data for downstream tasks, the approach remains contingent upon the base generative model's capability to produce viable initial samples. These considerations do not substantially compromise overall performance but indicate potential avenues for future enhancement.

## D Efficiency and Cost of Data Augmentation

To evaluate the computational efficiency of UTILGEN, we compare it against three representative generative augmentation methods: GIF [12], GAP [72], and DataDream [11]. The comparison is conducted under a unified setup using the ImageNet-1K dataset, where each method generates 1000 synthetic images per class. We decompose the computational pipeline into three stages: (1) *Model Optimization*, (2) *Policy Optimization*, and (3) *Image Generation*. UTILGEN incorporates feedback-driven optimization components at both the model and instance levels, introducing moderate overhead that remains manageable.

Table 14 presents the runtime taken for each stage, while Table 15 shows the peak GPU memory usage during the execution of each stage. All methods are evaluated on a multi-GPU server equipped with 8×A100 GPUs. Despite the feedback-based optimization, our framework remains computationally efficient. Overall, UTILGEN strikes a favorable balance between computational cost and data utility, demonstrating its scalability and practicality for real-world deployments.

## E Broader Impact

Our utility-driven augmentation approach facilitates more efficient model training while reducing reliance on real data, especially benefiting domains with limited or private datasets. By generating

Table 14: Computational cost (in hours) comparison on ImageNet-1K for generating 1000 images per class. Values are estimated averages.

| Method | Model Optimization | Policy Optimization | Image Generation |
|---|---|---|---|
| GIF [12] | – | – | ∼229.1h |
| GAP [72] | – | ∼45.1h | ∼93.7h |
| DataDream [11] | ∼10.8h | – | ∼40.2h |
| UTILGEN (Ours) | ∼7.7h | ∼25.1h | ∼41.6h |

Table 15: GPU memory usage (peak usage in GB) per stage. All values are measured during maximum workload per module on each GPU.

| Method | Model Optimization | Policy Optimization | Image Generation |
|---|---|---|---|
| GIF [12] | – | – | ∼25.9G |
| GAP [72] | – | ∼5.2G | ∼4.5G |
| DataDream [11] | ∼19.5G | – | ∼15.8G |
| UTILGEN (Ours) | ∼20.5G | ∼4.4G | ∼4.3G |

task-specific synthetic training data, it enhances learning efficiency and lowers dependence on large-scale real datasets. Nonetheless, since the synthesis process is guided by a small set of real images, the generated data may inadvertently inherit and amplify biases present in the original samples.

## F  Dataset Details

To evaluate UTILGEN's performance, we utilize eight benchmark datasets spanning a variety of classification tasks: coarse-grained classification, fine-grained classification, and texture classification.

The coarse-grained datasets include ImageNet-1k-Subset [73], ImageNet-100-Subset [73] and Caltech 101 [74]. ImageNet-1k-Subset [73] and ImageNet-100-Subset [73], both randomly sampled with 100 images per class. ImageNet-100-Subset is a subset of 100 animal-related classes from ImageNet-1K. Caltech 101 [74] consists of 101 object categories. For fine-grained classification, we use Oxford Pets [75], Food101-Subset [76], Flowers 102 [68], and CUB-200-2011 [77], with Food101-Subset [76] being a curated subset of Food101 [76] containing 101 food categories. Other datasets follow their original training and validation setups. Texture classification is evaluated using the DTD [78] dataset, which contains 47 texture categories. Detailed dataset statistics are provided in Table 16, summarizing the number of classes, training samples, and test samples for each dataset. It is important to note that datasets with a higher number of classes or fewer average samples per class present greater challenges in terms of classification and generalization.

Table 16: Statistics of the benchmark datasets

| Dataset | Task Type | Classes | Training Data | Test Data |
|---|---|---|---|---|
| ImageNet-1k-Subset [73] | Coarse-grained object classification | 1000 | 100,000 | 50,000 |
| ImageNet-100-Subset [73] | Coarse-grained object classification | 100 | 10,000 | 5,000 |
| Caltech 101 [74] | Coarse-grained object classification | 101 | 3060 | 6084 |
| Oxford Pets [75] | Fine-grained object classification | 37 | 3680 | 3669 |
| Food101-Subset [76] | Fine-grained object classification | 101 | 10100 | 25,250 |
| Flowers 102 [68] | Fine-grained object classification | 102 | 6,552 | 818 |
| CUB-200-2011 [77] | Fine-grained object classification | 200 | 5,994 | 5,794 |
| DTD [78] | Texture classification | 47 | 1880 | 1,880 |

