# OpenReview forum: "UtilGen: Utility-Centric Generative Data Augmentation with Dual-Level Task Adaptation"
_NeurIPS.cc/2025/Conference — NeurIPS 2025 poster_

### Official Review · Reviewer_TgCV · 2025-06-28

**Clarity:** 3
**Significance:** 3
**Originality:** 3
**Rating:** 4
**Confidence:** 4

**Summary:**

While most existing generative data augmentation methods prioritize the generation of high-fidelity and diverse images, this paper argues that practical tasks and model architectures often necessitate tailored training data distributions rather than generic visual quality. To address this, the authors propose Utility-Generated Data Augmentation (UtilGen), a framework that explicitly optimizes synthetic data to enhance task-specific performance, rather than merely adhering to visual standards. UtilGen introduces three key components: 1) Task-Oriented Data Valuation, which is a meta-learned weight allocation network that quantifies the task-specific utility of synthetic data. 2) Model-Level Generation Capability Optimization, which finetune the generator model (stable diffusion) using  DPO. 3) Instance-Level Generation Policy Optimization, which optimizes prompt embedding and input noise to maximize the utility weights of generated images. Extensive experiments across 8 datasets demonstrate that UtilGen outperforms prior state-of-the-art (SOTA) methods by an average of 3.49% in performance, validating its effectiveness in aligning synthetic data with downstream task requirements

**Questions:**

1.	How sensitive is the performance of UtilGen to the choice of the initial classifier architecture and training/val data selection?

2.	Are there any scenarios or types of datasets where use of UtilGen might not be effective?

3.	The framework seems to have several hyperparameters for each components. How sensitive is the method to the choice of these hyperparameters, and do you have any guidelines for setting them?

4.	How does this approach perform when we use pretrained vision transformer models as the classifier?

**Ethical Concerns:**

["NO or VERY MINOR ethics concerns only"]

**Final Justification:**

I appreciate the author's response to all my questions. After reviewing the rebuttal, I maintain that my original rating remains appropriate.

**Limitations:**

Yes, listed in appendix

**Quality:**

3

**Strengths And Weaknesses:**

Strength:

•	The paper's main strength is its proposal of a utility-centric paradigm for generative data augmentation. This is a departure from the conventional focus on visual fidelity and diversity, offering a more targeted and more effective approach to data augmentation.

•	A meta-learned weight allocation network provides an efficient, retraining-free solution for task-oriented data valuation.

•	Model-level and instance-level optimization provides a comprehensive framework for tailoring both the generative model and the individual generated samples for maximizing utility weights.

•	Paper provides rigorous experimental results on various datasets, showcasing an average of 3.49% performance improvement compared to previous SOTA approach

•	Paper provides all necessary ablation study and results which provide valuable insights into why the proposed method is effective.

Weakness:

•	The proposed framework's complexity arises from its integration of multiple modules and training stages (e.g., TODV, MLCO, ILPO, DPO, textual inversion), which may pose challenges for practical implementation

•	While the paper emphasizes the efficiency of its utility evaluation method, the overall framework, e DPO fine-tuning and instance-level optimization, could be computationally intensive. The computation cost is given in the appendix.

•	The performance of UtilGen is likely dependent on the quality of the initial classifier and the representativeness of its feature, as the utility weightage network uses the feature from the classifier model. The paper could have explored the sensitivity of overall method on choice of different classifier such as CNNs vs pretrained transformer models.

•	Paper could also compare their approach against simple augmentation or self-supervised pretrained models.

---

> ### Author Rebuttal · Authors · 2025-07-30
>
> Dear Reviewer,
>
> Thank you for your thoughtful feedback and insightful questions. Your comments have helped us to identify areas where we can improve the clarity and rigor of our paper. We have carefully considered your suggestions and provide our detailed responses below.
>
> >  **Q1. The proposed framework is complex due to the integration of multiple modules.(Weakness 1)**
>
> We thank the reviewer for this thoughtful concern. Each module in our framework is designed with clear modularity, allowing **independent development and evaluation**. This facilitates practical adoption while enabling the components to work synergistically for the substantial performance gains observed in our experiments.
>
> >  **Q2. Efficiency concerns regarding the overall computational cost of the framework.(Weakness 2)**
>
> We thank the reviewer for highlighting this important aspect. While UtilGen integrates several components to improve sample quality and task alignment, we would like to clarify that the overall framework remains computationally reasonable. As detailed in Appendix D, UtilGen exhibits favorable GPU memory usage and runtime compared to existing baselines. **We believe these costs are justified by the substantial accuracy improvements, and we plan to further explore efficiency optimizations in future work.**
>
> **Table: Computational cost (in hours) comparison on ImageNet-1K for generating 1000 images per class**
>
> | Method              | Model Optimization | Policy Optimization | Image Generation |
> |:--------------------|:------------------:|:-------------------:|:----------------:|
> | GIF[2]                 | --                 | --                  | ~229.1h          |
> | GAP[1]                 | --                 | ~45.1h              | ~93.7h           |
> | DataDream[1]           | ~10.8h             | --                  | ~40.2h           |
> | UtilGen       | ~7.7h              | ~25.1h              | ~41.6h           |
>
> **Table: GPU memory usage (peak usage in GB) per stage**
>
> | Method              | Model Optimization | Policy Optimization | Image Generation |
> |:--------------------|:------------------:|:-------------------:|:----------------:|
> | GIF[2]                 | --                 | --                  | ~25.9G           |
> | GAP[1]                 | --                 | ~5.2G               | ~4.5G            |
> | DataDream[3]           | ~19.5G             | --                  | ~15.8G           |
> | UtilGen     | ~20.5G             | ~4.4G               | ~4.3G            |
>
>
>
> >  **Q3. The sensitivity of overall method on choice of different classifier such as CNNs vs pretrained transformer models.(Weakness 3)**
>
> We thank the reviewer for this valuable point. To investigate the sensitivity of UtilGen to the choice of classifier, we conducted experiments on ImageNet-100 using various architectures, including CNNs (ResNeXt, WideResNet, MobileNet) and pretrained transformer models (CLIP). As shown in the table below, UtilGen consistently improves downstream performance across different classifiers, **demonstrating robustness to the classifier choice**. This validates that our weight network can effectively leverage diverse model architectures without being overly dependent on a specific backbone.
>
> | Method      | ResNeXt-50 | WideResNet-50 | MobileNetV2 | CLIP   |
> |-------------|:----------:|:-------------:|:-----------:|:------:|
> | GIF         | 27.54%     | 27.84%        | 31.24%      | 69.24% |
> | GAP         | 27.66%     | 27.76%        | 32.72%      | 69.62% |
> | DataDream   | 31.24%     | 31.76%        | 35.48%      | 71.14% |
> | **UtilGen** | **37.50%** | **37.78%**     | **40.56%**  | **72.18%** |
>
> >  **Q4. Compare the approach against simple augmentation or self-supervised pretrained models.(Weakness 4)**
>
> We thank the reviewer for the helpful suggestion. We compare UtilGen with Cutout[4] and CLIP on ImageNet-100 using a 10× enlarged training set that combines both real and synthetic data. For fair comparison, Cutout[4] and UtilGen are trained using ResNet-50. UtilGen significantly outperforms Cutout[4] and even slightly surpasses CLIP in the zero-shot setting, while fine-tuning CLIP on the same augmented dataset yields the best performance. These results **highlight the effectiveness of our approach compared to simple augmentation and self-supervised pretrained models**.
>
> | Method                                              | ImageNet-100 Accuracy (\%) |
> |-----------------------------------------------------|:---------------------------:|
> | Cutout[4] (ResNet-50)                              |           40.32            |
> | CLIP (Zero-Shot)                                    |           69.12            |
> | UtilGen (ResNet-50)                        |           69.74            |
> | CLIP (Fine-tuned on real + synthetic data, 10×)     |         74.68          |
>
>
> >  **Q5. Sensitivity to the choice of initial classifier architecture and train/val data selection.(Question 1)**
>
> As noted in **Q3**, UtilGen is not sensitive to the initial classifier architecture, showing consistent gains across different backbones. It is also robust to variations in training/validation data selection: for training the weight network, we randomly sampled a few real images per class, and observed consistent improvements across different random splits. This stability indicates that UtilGen is not sensitive to the choice of train/val data.
>
>
> >  **Q6. Potential scenarios or dataset types where UtilGen may not perform well.(Question 2)**
>
> Since UtilGen relies on a few real images for guidance, **very low-quality inputs** (e.g., noisy or low-resolution) may reduce generation quality by affecting textual inversion. Nonetheless, this has minimal impact on overall performance and does not compromise our main contributions. This also suggests a valuable direction for future work—improving robustness to low-quality real data.
>
> >  **Q7. Sensitivity to hyperparameter choices and guidelines for tuning.(Question 3)**
>
> We appreciate the reviewer’s question. The implementation details and hyperparameter configurations for each component of our framework are provided in Appendix B.
>
> Among all components, the **most sensitive hyperparameter** is the **guidance scale** used in the diffusion model for image generation. We conducted experiments to study its impact:
>
> | Guidance Scale | ImageNet-100 Accuracy (%)|
> |----------------|:------------------:|
> | 5              | 37.20                |
> | 2              | 40.86                |
>
>
> >  **Q8. Performance when using pretrained vision transformer models as the classifier.(Question 4)**
>
> We evaluated UtilGen using CLIP as the downstream classifier and compared it with DataDream under the same setting on the ImageNet-100 dataset. Both methods use only 5× synthetic data for training. UtilGen consistently achieves better performance:
>
> | Method                 | ImageNet-100 Accuracy (%) |
> |------------------------|:------------------------:|
> | DataDream (CLIP)       |           71.14          |
> | UtilGen (CLIP)         |       **72.18 (+1.04)**  |
>
> This shows that UtilGen remains effective even when using pretrained transformer-based classifiers.
>
> **References**
>
> [1] Controlled Training Data Generation with Diffusion Models
>
> [2] Expanding Small-Scale Datasets with Guided Imagination
>
> [3] DataDream: Few-shot Guided Dataset Generation
>
> [4] Improved regularization of convolutional neural networks with cutout.
>
> Thank you once again for your valuable feedback, which has helped us to significantly improve the scientific rigor of our paper. We will incorporate the corresponding revisions and clarifications into the updated manuscript. If you have any further questions or concerns, we would be glad to address them.
>
> Sincerely,
>
> The Authors

---

### Official Review · Reviewer_4Eig · 2025-07-01

**Clarity:** 2
**Significance:** 3
**Originality:** 3
**Rating:** 4
**Confidence:** 4

**Summary:**

This paper addresses the problem of designing data-generation models that are well aligned with a downstream task. A key element of this setup is a weight network, which learns to define weights associated with data samples, reflective of their utility for a given task, e.g., classification. Once this weight network is learned, it is used to refine the synthetic data generation, such that it yields high-weight data. Emphasis is placed on diffusion models. Additionally, the noise generation and text-prompt embeddings are also optimized, again leveraging the weight network.

**Questions:**

1. I found the discussion in Sec 3.1 not very clear, particularly Eqs (2) and (3). In (3), you are optimizing \phi, it seems, parameters associated with the weight network. It seems therefore that you are optimizing \omega_i. These weights impact the learning of \theta in (2). Please provide some clarification of what you are saying mathematically here, as it is not very clear. I was helped by Fig 2, which at least was able to give me the idea of what you are trying to say.

2. Typos:

a) Line 114: "preferences preferences"
b) Line 119: "during training process"; missing a "the"
c) Line 146: "MLPO"; I think you mean MLCO

**Ethical Concerns:**

["NO or VERY MINOR ethics concerns only"]

**Final Justification:**

There were technical issues in the original submission that were not clear as written, and this led to excessive reliance on figures by the reader. The authors did a nice job in their rebuttal of addressing these concerns, and I am confident that they will address these issues in the revision. This is a borderline paper, but I recommend acceptance.

**Limitations:**

Yes

**Paper Formatting Concerns:**

No issues

**Quality:**

3

**Strengths And Weaknesses:**

Strengths:

1. The paper addresses an important problem and provides an effective procedure
2. The demonstration figures are helpful in giving intuition (Fig 1) and in providing a concise summary of the approach (Fig 2)

Weaknesses:

1. Aspects of the paper are not very clearly written (Fig 2 was important for understanding the idea, as some of the text description could have been more clearly written)
2. It is said that an advantage of this approach is that it doesn't need classifier-based methods to evaluate the quality of the generated data. However, the method used to learn the weight network relies heavily on training aligned with a downstream task, e.g., classification. There is a need for sufficient training and validation data to make this model (weight network) effective. While it is true that the later stages of the setup do not need a classifier, the entire approach rests on the quality of the weight network (stage 1), and a lot of data may be needed to learn that well.

---

> ### Author Rebuttal · Authors · 2025-07-30
>
> Dear Reviewer,
>
> Thank you for your thoughtful feedback and insightful questions. Your comments have helped us to identify areas where we can improve the clarity and rigor of our paper. We have carefully considered your suggestions and provide our detailed responses below.
>
> >  **Q1. Concern regarding the claim of not requiring a classifier-based method to evaluate data utility, and the potential reliance on training and validation data for utility learning (Weakness 2)**
>
> We appreciate these insightful questions and would like to provide further clarification.
>
> **- Evaluation without exhaustive classifier-based re-training**: We appreciate the reviewer’s comment and would like to clarify that our claim of not requiring classifier-based methods refers to **avoiding exhaustive re-training of downstream classifiers**. Instead, it uses a weight network as a surrogate utility estimator, which enables efficient evaluation of synthetic samples based on their estimated contribution to downstream performance. This avoids the computational burden associated with repeatedly training classifiers on different subsets of generated data.
>
> **- Training data requirements of the weight network**: We thank the reviewer for raising this important point. While the weight network is indeed trained on a large number of samples, we would like to clarify that **this does not require a large-scale real labeled dataset**. In fact, **UtilGen naturally supports few-shot learning**, requiring only minimal labeled data while maintaining competitive performance. The remaining training data for the weight network consists entirely of synthetic samples. This capability is empirically validated in our comparative experiments with DataDream[4] on ImageNet-100, where UtilGen demonstrates superior effectiveness in data-scarce regimes.
>
> **Experimental Setup**:
> - Half of the images are simultaneously used for:
>   - The training set for Weight-Net training
>   - Textual inversion optimization
> - The remaining half serves as the validation set for Weight-Net training.
>
> | Method    | 2-shot (%) | 4-shot (%) | 8-shot (%) | 16-shot (%) |
> |:---------:|:----------:|:----------:|:----------:|:-----------:|
> | DataDream[1] | 21.70     | 27.18      | 33.44      | 35.48      |
> | UtilGen   | 22.32      | 29.16      | 36.82      | 39.62       |
>
> > **Q2. The Clarity of Section 3.1 and Equations (Weakness 1 & Question 1)**
>
> We thank the reviewer for the helpful suggestion, and we will provide a more transparent and detailed description of the equations in Section 3.1 to improve clarity and understanding. Below we provide a step-by-step clarification of the interaction between the classifier parameters $\theta$, the weight network parameters $\phi$, and the computed weights $\omega_i$.
>
> ---
>
> ### **Bi-Level Optimization Procedure**
>
> Our method employs a bi-level optimization framework with two alternating stages. The goal is to meta-learn a weight network $\mathcal{W}_\phi$ that assigns utility scores $\omega_i \in [0, 1]$ to data points, such that samples with higher utility contribute more effectively to downstream generalization.
>
> **Stage 1: Inner Optimization — Classifier Training**
>
> Given a current weight network $\mathcal{W}_\phi$, we first compute the weight $\omega_i$ for each training example using the weight network.
>
> The classifier parameters $\theta$ are then optimized on the merged training set using a weighted loss:
>
> $$
> \theta^*(\phi) = \arg\min_{\theta} \frac{1}{N} \sum_{i=1}^{N} \omega_i \cdot \mathcal{L}(f(x_i; \theta), y_i)
> \quad \text{(Eq. 2)}
> $$
>
> Note that $\theta^*$ is a function of $\phi$ because the weights $\omega_i$ used in training depend on $\mathcal{W}_\phi$. This shows that the optimal classifier $\theta^{\ast}(\phi)$ is implicitly determined by the weight network parameters $\phi$ via the computed weights.
>
> **Stage 2: Outer Optimization — Weight Network Update**
>
> Once the classifier has been trained with the current weights $\omega_i$, we evaluate its generalization on a validation set. The goal of the outer optimization is to update $\phi$ such that the classifier trained with weighted loss performs well on validation set:
>
> $$
> \phi^{\ast}(\theta) = \arg\min_{\phi} \frac{1}{M} \sum_{j=1}^{M} \mathcal{L}(f(x_j; \theta^{\ast}(\phi)), y_j)
> \quad \text{(Eq. 3)}
> $$
> This outer loop updates $\phi$ so that the next round of training with newly computed weights $\omega_i$ better guide the classifier training, leading to improved downstream generalization.
>
> This process is repeated iteratively: the weight network is progressively refined to better estimate the utility of each sample, thereby guiding the classifier to train more effectively using weighted loss in the next iteration. Over time, the weight network progressively learns to assign higher weights to training samples that contribute more significantly to improved validation performance. In other words, the learned weights act as a surrogate for each sample's utility with respect to downstream generalization.
>
> ---
>
> ### **Intuition**
>
> In essence, we are **meta-learning** the weight network to simulate the effect of each training sample on validation performance:
>
> - If increasing the weight of sample $x_i$ leads to a lower validation loss, $\mathcal{W}_\phi$ is encouraged to assign it a higher weight.
> - Conversely, samples with low or negative impact on generalization are down-weighted.
>
> This interaction forms a feedback loop between the classifier training (inner loop) and the weight assignment (outer loop), enabling the weight network to serve as a proxy utility estimator for guiding data generation in later stages.
>
> We appreciate the reviewer’s reference to Fig. 2 — we will further revise Section 3.1 to explicitly describe these dependencies and clarify the semantics of the equations involved.
>
>
> >  **Q3. Typos and Wording Issues.(Question 2)**
>
> We thank the reviewer for pointing out the typos and wording issues. We have carefully revised the manuscript to correct these and other textual errors for improved clarity and accuracy.
>
> **References**
>
> [1] DataDream: Few-shot Guided Dataset Generation
>
> Thank you once again for your valuable feedback, which has helped us to significantly improve the scientific rigor of our paper. We will incorporate the corresponding revisions and clarifications into the updated manuscript. If you have any further questions or concerns, we would be glad to address them.
>
> Sincerely,
>
> The Authors

---

> > ### Comment · Reviewer_4Eig · 2025-08-05
> >
> > Thank you for your thoughtful response. I will leave my already-high score as is.

---

> > > ### Author Response · Authors · 2025-08-06
> > >
> > > We would like to express our sincere gratitude to the reviewer once again for your valuable time and constructive feedback.

---

### Official Review · Reviewer_2gVY · 2025-07-03

**Clarity:** 3
**Significance:** 3
**Originality:** 4
**Rating:** 5
**Confidence:** 4

**Summary:**

This paper presents a data augmentation technique that uses synthetic samples from a diffusion model. The authors introduce a task dependent objective for sample generation. First they formulate a bilevel optimization problem to train a simple MLP that assigns weights to samples in the cross entropy loss, with the MLP computing a utility gain for each sample. Next they leverage this utility network to generate negative and positive samples for direct preference alignment. They also propose model level and instance level tuning of class specific prompt embeddings and initial noise for each generated sample based on the utility MLP. Experimental results show improved performance when a target model is trained on synthetic data alone and on a mix of real and synthetic data using the proposed framework.

**Questions:**

I also have the following questions and requests:

1. I am curious about the performance gap of UtilGen compared with other baselines under different synthesis budgets. How does the gap change as more data is augmented?

2. Why not first perform textual inversion to adapt the diffusion model to each class and then use the synthesized samples via the textual inversion (concatenated to the real data) to train the utility MLP in the first step? This would reduce distribution shift for the MLP when later inferred for diffusion alignment and DPO labeling.

3. I would appreciate if the authors could explain how sample weights relate to the model’s cross entropy loss. Is there a trend between sample weights and the uncertainty of the $ \theta $ model, and if so how does this trend evolve across different optimization stage of $\theta$?

4. Following on the previous point, although it may be outside the current method’s scope and that is absolutely fine if it is not addressed because of time constraints, it would be interesting to evaluate performance when the utility MLP is trained iteratively:

   1. First augment the dataset by a factor of 1×.

   2. Fine tune the MLP to produce updated sample weights after training on this augmented set.

   3. Synthesize new samples using the improved MLP to achieve an 2× augmentation factor and so on to 3x and beyond...

   This experiment would reveal whether adapting the MLP for weighting at different training stages yields additional gains.


I thank the authors for the good work and I would be willing to raise my score upon addressing the questions.

**Ethical Concerns:**

["NO or VERY MINOR ethics concerns only"]

**Final Justification:**

The concerns regarding the lack of additional experimental details on solving the bilevel optimization problem and textual inversion have been resolved. The authors have also provided additional ablations that further clarify the effectiveness of their method, including comparisons against baselines under different data regimes, mixing text-inverted synthetic data with real data for training the MLP, and a diversity analysis.

I have raised my score accordingly.

**Limitations:**

yes

**Quality:**

4

**Strengths And Weaknesses:**

**Strengths**

• The paper is well structured; topics flow logically.
• Formatting is clean; figures and tables are of high quality.
• The proposed method is novel and improves over prior work.
• Ablations and experimental design follow a logical sequence and address readers’ questions as they arise.

**Weaknesses**

1. The implementation details require further clarification on several points of the method:

   a. How is the bilevel optimization solved? Obtaining $\theta^*(\phi)$ requires gradient unrolling to backpropagate through $\phi$, which can demand substantial memory.

   b. How is the prompt optimized during the denoising stage? Diffusion models involve multiple denoising steps and backpropagating through prompt parameters may incur high computational cost.

2. The authors claim that UtilGen enhances diversity but the connection between the utility MLP and diversity promotion is not clear. It would be useful to report a diversity metric such as pairwise distance in feature space (or any other) to quantify the diversity of synthesized samples.

---

> ### Author Rebuttal · Authors · 2025-07-30
>
> Dear Reviewer,
>
> We sincerely appreciate your thoughtful feedback and insightful questions, which have helped identify key areas for improving our paper's clarity and academic rigor. After careful consideration of your suggestions, we provide the following detailed responses.
>
> >  **Q1. How is the bilevel optimization solved?(Weakness 1.a)**
>
> We implement the bi-level optimization through **one-step gradient unrolling**, where the classifier performs a single gradient update step on the training data, and then the weight network is updated based on the validation loss. While gradient unrolling can be memory-intensive, in our setup it remains computationally tractable due to the lightweight MLP architecture of the weight network and the limited unrolling steps. This approach effectively avoids excessive computational and time costs, making the optimization practical and efficient.
> Below, we report the peak GPU memory usage during training with and without the weight network on ResNet-50 as the classifier:
>
> | Classifier  | Weight Network | Peak Memory (GB) |
> |-------------|:---------:|:---------:|
> | ResNet-50   | ✗              | 14.8              |
> | ResNet-50   | ✓              | 22.9              |
>
> As shown, incorporating the weight network introduces only a modest increase in memory consumption, demonstrating the practicality of our bilevel optimization implementation.
>
> >  **Q2. How is the prompt optimized during the denoising stage?(Weakness 1.b)**
>
> We thank the reviewer for raising this important point. In UtilGen, we optimize the prompt embeddings by backpropagating through the denoising process of the diffusion model. This involves computing gradients through denoising steps, which can indeed incur additional computational overhead. We acknowledge this as a limitation of our current design. However, in practice, **the overhead remains manageable because we only optimize one prompt per class**, rather than per instance. This significantly reduces the total optimization cost and makes the method practically feasible.
>
> To evaluate the computational efficiency, we compare our method UtilGen with GAP[1] in terms of memory and runtime for prompt optimization. A broader comparison of UtilGen and all other baselines in terms of GPU memory usage and runtime is provided in Appendix D.  In future work, we plan to explore more efficient prompt optimization strategies to further reduce the computational burden while preserving performance.
>
> | Method     | GPU Memory (GB) | Runtime (GPU hours) | ImageNet-100 Acc (%) |
> |------------|:---------:|:---------:|:---------:|
> | GAP[1]        | 5.2             | 36.1             | 30.94    |
> | UtilGen    | 4.4             | 20.1             | 40.86   |
>
> >  **Q3. Diversity analysis of synthetic data.(Weakness 2)**
>
> We thank the reviewer for the valuable suggestion. UtilGen maintains data diversity during the optimization of data generation. To quantitatively evaluate the diversity of synthesized samples, we compare the **mean intra-class diversity** of synthetic data produced by Stable Diffusion v2.1, DataDream[3], and UtilGen on ImageNet-100.
>
> The diversity metric is defined as the mean pairwise cosine distance between CLIP (ViT-L/14) features within each class, averaged across all classes. Higher values indicate greater intra-class diversity, which typically benefits model generalization.
>
> | Method                 | Mean Intra-Class Diversity ↑ |
> |------------------------|:----------------------------:|
> | Stable Diffusion v2.1   | 0.5815                       |
> | DataDream [3]           | 0.5238                       |
> | UtilGen         | 0.6054                       |
>
> These results demonstrate that UtilGen produces diverse samples within classes compared to the baselines, supporting the claim that our utility-guided generation process also does not harm or suppress diversity, which is beneficial for downstream tasks.
>
> >  **Q4. How does the performance gap between UtilGen and other baselines evolve under different synthesis budgets?(Question 1)**
>
> We thank the reviewer for the insightful question. We conducted a controlled comparison of UtilGen and four baselines (SD v2.1, GIF[2], GAP[1], and DataDream[3]) under varying synthesis budgets (i.e., the ratio of synthetic to real data). As shown in the table below, UtilGen consistently outperforms all baselines, and the performance gap becomes more pronounced as the synthesis budget increases.
>
> This trend suggests that **UtilGen better scales with larger amounts of generated data**, owing to its utility-centric data augmentation.
>
> | Synthesis Budget (× real data) | SD v2.1 | GIF[2]  | GAP[1]  | DataDream[3] | **UtilGen (Ours)** |
> |---------------------------------|:---------:|:------:|:------:|:-----------:|:--------------------:|
> | 1×                            |  12.18   | 13.44 | 15.02 | 17.12      | 18.04 **(+0.92)**           |
> | 3×                            |  20.20 | 21.90 | 21.16 | 25.26      | 28.52 **(+3.26)**           |
> | 5×                            |  27.96   | 31.94 | 30.94 | 35.48      | 40.86 **(+5.38)**           |
>
>
>
>
> >  **Q5. Why not perform textual inversion before utility learning to reduce distribution shift?(Question 2)**
>
> We thank the reviewer for the insightful suggestion. In our initial design, we trained the utility MLP using synthetic samples directly generated from the pre-trained SD v2.1 model, without applying textual inversion. This was motivated by the observation that such raw generations often contain low-utility samples (e.g., off-distribution or task-irrelevant data), which we believed would help the utility MLP better distinguish varying levels of usefulness.
>
> Following the reviewer’s suggestion, we conducted additional experiments using class-adapted samples generated via textual inversion to train the utility MLP. These adapted samples reduced the distribution shift encountered during downstream diffusion alignment and DPO labeling. Our experiments confirmed that using these samples indeed improves the effectiveness of the utility MLP.
>
> Moreover, we also found that **combining both types of data—raw samples from SD v2.1 and class-adapted samples from textual inversion yields the best overall performance**. This hybrid training strategy not only mitigates distribution mismatch but also enhances the MLP’s ability to assess utility across a broader quality spectrum.
>
> | Training Strategy for Utility MLP     | ImageNet-100  Accuracy (%) |
> |:------------------------------------- |:-------------------------------:|
> | Raw SD v2.1 samples only + real data              |              40.86              |
> | Textual inversion samples only + real data       |              41.58              |
> | Raw + Textual inversion (hybrid) + real data     |            **41.94**            |
>
> >  **Q6. What is the relationship between sample weights and model uncertainty or loss?(Question 3)**
>
> In our analysis, we observe a nonlinear relationship between the sample weights predicted by the utility MLP and the model’s cross-entropy loss. While we cannot include plots here, the general trend roughly forms an **“inverted U-shaped” curve**, where the weights tend to increase with the loss when the loss is relatively small, but starts to decrease as the loss becomes large.
>
> This pattern aligns well with intuition: samples with very low loss are often easy or redundant, thus contributing less utility to model performance; samples with very high loss are likely noisy or exhibit significant distribution shifts, and are therefore also assigned lower weights.
>
> Regarding the evolution of this relationship across different optimization stages, our experiments indicate that the correlation between sample weights and cross-entropy loss gradually stabilizes to this nonlinear “inverted U-shaped” trend as training progresses.
>
>
> >  **Q7. Would iterative utility learning improve performance?(Question 4)**
>
> We thank the reviewer for this insightful suggestion. We have conducted experiments to evaluate the iterative training of the utility MLP as proposed. Specifically, we compared two settings:
>
> 1. **Direct 5× augmentation:** generating all synthetic samples at once using a single utility MLP trained initially.
>
> 2. **Iterative augmentation:** first generating 1× synthetic data, fine-tuning the utility MLP based on the augmented set, then generating an additional 1× data with the improved MLP, and repeating this process until reaching 5× augmentation.
>
> Our results show that **the iterative approach achieves better performance compared to direct generation**, confirming that adapting the utility MLP at different training stages can bring additional gains. However, this improvement comes at the cost of increased computational overhead due to repeated MLP fine-tuning and data generation steps.
>
> | Augmentation Strategy     | Accuracy (%) |
> |:--------------------------|:------------:|
> | Direct 5× Augmentation    |     40.86    |
> | Iterative (1× → 2× → 5×) |  43.10 **(+2.24)**   |
>
> **References**
>
> [1] Controlled Training Data Generation with Diffusion Models
>
> [2] Expanding Small-Scale Datasets with Guided Imagination
>
> [3] DataDream: Few-shot Guided Dataset Generation
>
> Thank you once again for your valuable feedback, which has helped us to significantly improve the scientific rigor of our paper. We will incorporate the corresponding revisions and clarifications into the updated manuscript. If you have any further questions or concerns, we would be glad to address them.
>
> Sincerely,
>
> The Authors

---

> > ### Comment · Reviewer_2gVY · 2025-08-05
> >
> > I thank the authors for conducting the requested experiments and clarifying the computational overhead of their method. Including the experimental details on bilevel optimization and textual inversion in the main text improves the clarity of the work, and adding the additional experiments to the appendix will strengthen the overall presentation. I also believe the diversity result is important to include in the paper, as it directly supports the authors’ claim that their scheme promotes diversity.
> >
> > Overall, I believe the authors have thoroughly addressed all my concerns, and I will raise my score accordingly.

---

> > > ### Author Response · Authors · 2025-08-05
> > >
> > > We're pleased to hear that our response has addressed your concerns. Thank you once again for your time and insightful feedback. We believe that your suggestions and the enhanced experimental results have significantly strengthened our work and will contribute meaningfully to the community.

---

### Official Review · Reviewer_or1d · 2025-07-03

**Clarity:** 4
**Significance:** 3
**Originality:** 2
**Rating:** 4
**Confidence:** 4

**Summary:**

The paper introduces a utility-centric data augmentation framework aimed at improving the task-specific effectiveness of synthetic data used in computer vision models. While existing data augmentation methods using generative models primarily focus on visual fidelity and diversity, UTILGEN shifts the paradigm to focus on task-specific utility—how useful synthetic data is for enhancing downstream task performance.

To achieve this, UTILGEN employs a meta-learned weight allocation network that efficiently evaluates the task-specific utility of each synthetic data sample without requiring full training cycles. These utility scores then guide a dual-level optimization process: 1. Model-Level Optimization: Adapts the generative model itself to the downstream task using methods like Direct Preference Optimization (DPO).
2. Instance-Level Optimization: Fine-tunes generation policies (e.g., prompt embeddings, initial noise) to produce more useful data samples.

Their framework is evaluated across **eight** benchmark datasets and shows an average improvement of 3.49% in accuracy over state-of-the-art augmentation methods. Notably, a ResNet-50 trained solely on UTILGEN-generated synthetic data outperforms its real-data-trained counterpart on multiple tasks. The framework is also effective across different architectures (e.g., ResNeXt, WideResNet, MobileNet), demonstrating versatility and robustness.

**Questions:**

The 1-5 questions asked in the Weaknesses section, would certainly affect the evaluation scores.

**Ethical Concerns:**

["NO or VERY MINOR ethics concerns only"]

**Limitations:**

The missing limitation related to *Limited Reusability of Synthetic Data* is discussed under the Weakness section above. Apart from this, there are no other major weaknesses.

**Paper Formatting Concerns:**

No concerns.

**Quality:**

3

**Strengths And Weaknesses:**

**Strengths**:

1.  Paper is very well written and easy to understand.

2. UTILGEN introduces a novel shift in data augmentation—from optimizing visual fidelity and diversity to maximizing task-specific utility. This addresses a major gap in generative augmentation by aligning synthetic data generation with actual model learning needs.

3. The "meta-learned weight allocation network" estimates task-specific sample utility without full retraining. Combined with influence function analysis and PCA-based visualization, the framework demonstrates that it generates data with stronger positive impact and clearer alignment within high-utility region. Such analysis is missing in prior methods.

4. UTILGEN’s synthetic data scales effectively—performance continues to improve from 1× to 5× synthetic data, with increasing positive influence on downstream models.

5. The experiments with model variants is extensive, making the results robust and effective across different multiple architectures (ResNet, ResNeXt, WideResNet, MobileNetV2).

**Weaknesses**:

1. The meta-learned weight network provides an efficient way to estimate task-specific utility, but its internal logic remains unclear. While data influence and distribution analyses validate its effectiveness, understanding why certain samples are rated as high-utility is not directly interpretable to user? More insights on this will aid the method's effectiveness and user's understanding.

2. While UTILGEN introduces a task-aware, feedback-driven approach to data generation, it does not compare against DisCL [1], a method that also incorporates task-adaptive feedback—albeit through a curriculum learning framework. Both methods share the goal of improving downstream performance by guiding data synthesis based on task-specific signals. However, DisCL is absent from both the related work discussion and experimental baselines. Including such a comparison is important to properly position UTILGEN within the broader landscape of task-adaptive data generation methods, and to validate its generality and superiority over other task-aware strategies.

3. Although UTILGEN avoids full retraining cycles via its meta-learned utility predictor, the overall framework—including dual-level optimization (model- and instance-level) and task feedback loops—remains computationally intensive. The cost-benefit tradeoff compared to collecting more *labeled real data* **isn’t fully analyzed** in the paper.

4. Their results shows that synthetic data alone can match or surpass real-data-trained models on multiple benchmarks.
However, This seems to differ from findings in prior work such as DataDream [2], where synthetic data is shown to be most effective when mixed with real data. Could the authors elaborate on what aspects of UTILGEN (e.g., the utility-guided generation process) you believe contribute most to this breakthrough in synthetic-only training performance?

5. *Limited Reusability of Synthetic Data*: The framework still requires fine-tuning for each downstream task, including model-level generation optimization and instance-level controls. This limits out-of-the-box reusability of synthetic data across tasks. How would authors address or acknowledge this limitation ? Also Can this framework be extended to few shot setting ?

**References**

[1] Diffusion Curriculum: Synthetic-to-Real Generative Curriculum Learning via Image-Guided Diffusion.

[2] DataDream: Few-shot Guided Dataset Generation

---

> ### Author Rebuttal · Authors · 2025-07-30
>
> Dear Reviewer,
>
> We sincerely appreciate your thoughtful feedback and insightful questions, which have helped identify key areas for improving our paper's clarity and academic rigor. After careful consideration of your suggestions, we provide the following detailed responses.
>
> >  **Q1. Concern on the interpretability of the utility weight network and why certain samples are rated as high-utility.(Weakness 1)**
>
> Thank you for highlighting the need for further clarification regarding why certain samples are rated as high-utility. As mentioned in our paper, the weights learned by our network effectively reflect sample utility for downstream tasks. This behavior is theoretically grounded in the Meta-Weight-Net[1] framework, where the updating equation for Weight-Net parameters can be explained by that **samples that better comply with the knowledge derived from validation data (i.e., those that help improve the classification model's performance on downstream tasks) will have their weights increased**, while harmful samples are down-weighted. Although we cannot show the full derivation here due to space constraints, Meta-Weight-Net[1] provides rigorous proofs (Section 2.3 and Appendices B–C). Essentially, our weight allocation network inherits this property, where higher weights reliably indicate samples that enhance classification performance, thus guiding effective utility optimization. We will include further explanation in the revised version to improve clarity.
>
> > **Q2. Suggest including DisCL[2] as a task-adaptive baseline for comparison.(Weakness 2)**
>
> Thank you for the valuable suggestion. We have now included a comparison between UtilGen and DisCL[2] on the ImageNet-100 dataset. In addition, our main paper also presents a comparison with GAP[3], a feedback-driven approach that focuses on generating hard samples for downstream tasks. This unified comparison allows us to evaluate three different feedback-driven generation strategies. The evaluation covers two settings: synthetic data only, and a mixed setting combining synthetic and real data. The results confirm the superiority of our utility-guided data generation method compared to other baselines that are also guided by downstream task feedback.
>
> | Method     | Setting                  | ImageNet-100 Accuracy (%) |
> |------|:--------:|:-----------:|
> | GAP[3]        | Synthetic Only (5×)      | 30.94                     |
> | DisCL[2]  | Synthetic Only (5×)      | 37.82                     |
> | UtilGen    | Synthetic Only (5×)      | 40.86 (**+3.04**)         |
> | GAP[3]        | Synthetic (5×) + Real    | 53.14                     |
> | DisCL[2]  | Synthetic (5×) + Real    | 56.43                     |
> | UtilGen    | Synthetic (5×) + Real    | 61.14 (**+3.71**)         |
>
> We will include additional experimental comparisons and discussion with DisCL[2] in the revised manuscript to better position UtilGen among task-adaptive generation methods.
>
> > **Q3. Analysis of cost-benefit tradeoff compared to collecting more labeled real data.(Weakness 3)**
>
> We thank the reviewer for raising this important point regarding the cost-effectiveness of our approach. Compared to manually collecting and annotating real data, UtilGen significantly reduces both time and cost for dataset expansion. Specifically, manually annotating 10,000 images (e.g., collecting 100 images for each of 100 classes), according to Masterpiece Group, would typically take around 2 weeks and cost approximately \$800. In contrast, generating the same amount of data with UtilGen takes only about **0.94 hours** and costs roughly **\$20** for renting 8 V100 GPUs according to Google Cloud. Furthermore, on the ImageNet-100-Subset dataset (which contains 100 classes with 100 real images per class), we achieve better performance than the real-data baseline by using only 5× synthetic data. Generating this 5× synthetic dataset requires approximately **4.70 hours**, costing around **\$100** for renting 8 V100 GPUs.
>
> | Method        | Images | Time Required | Cost  | ImageNet-100 Accuracy(%)  |
> |---------|:-------:|:-------------:|:----------:|:--------:|
> | Manual Annotation | 10,000  | ~2 weeks      | \$800       | 38.58         |
> | UtilGen (1×)      | 10,000  | ~0.94 hours    | \$20        | 18.04       |
> | UtilGen (5×)      | 50,000  | ~4.70 hours    | \$100        | **40.86** |
>
> > **Q4. What enables UtilGen to achieve strong performance in the synthetic-only setting, unlike prior work?(Weakness 4)**
>
> We think this breakthrough in synthetic-only training stems from  shifting the optimization objective of synthetic data toward enhancing utility for downstream tasks, thereby **aligning the generative model’s optimization objective with the downstream task goals**.
>
> Previous methods primarily focus on enhancing the visual quality of synthetic images; however, there often exists a gap between generating visually high-quality data and producing data that effectively improves downstream model performance. In particular, many approaches struggle to directly optimize the utility of synthetic data for target tasks.
>
> On the other hand, efforts that concentrate on **optimizing intrinsic data attributes may introduce new limitations**. For example, DataDream[4] aims to enhance image fidelity and better align synthetic samples with the real data distribution. While effective in certain respects, this process can inadvertently reduce the diversity of the generated data. To quantify this, we compare the intra-class diversity of synthetic data generated by Stable Diffusion v2.1, DataDream[4], and UtilGen on ImageNet-100.The diversity metric is defined as the mean pairwise cosine distance between CLIP features within each class, averaged across all classes. Higher values indicate greater intra-class diversity.
>
>
> | Method                  | Mean Intra-Class Diversity ↑ |
> |------------|:-------------:|
> | Stable Diffusion v2.1  | 0.5815                      |
> | DataDream[4]          | 0.5238                       |
> | UtilGen (Ours)         | 0.6054                       |
>
>
> > **Q5. Limited reusability of synthetic data across tasks.(Weakness 5)**
>
> Thank you for raising this important point. While we evaluated UtilGen's generalization across network architectures, we acknowledge that the paper did not explicitly analyze data reusability across broader task settings. Below, we offer additional analysis and discussion regarding this aspect.
>
> If “different tasks” refers to directly using the generated data out-of-the-box as training data for different downstream models (e.g., other classifiers), we clarify that UtilGen-generated data demonstrates strong **cross-architecture transferability**. As evidenced by the table below, even if the utility predictor(i.e., Weight-Net) is trained using ResNet-50, the resulting data still yields strong performance when used with MobileNet and CLIP—suggesting that high-utility samples identified by the weight network are not tightly coupled to a specific architecture.
>
> | Classifier Used to Train Weight-Net | Downstream Model Using Synthetic Data |   Method   | ImageNet-100 Acc (%) |
> |:----------|:------:|:----------:|:---------:|
> | --                                  | WideResNet                            | DataDream[4]  |        31.76        |
> | ResNet-50                           | WideResNet                            |  UtilGen   |        36.40 **(+4.64)**        |
> | --                                  | CLIP                                  | DataDream[4]  |        71.42         |
> | ResNet-50                           | CLIP                                  |  UtilGen   |        72.14 **(+0.72)**        |
>
> However, when task differences involve changes in dataset or domain, UtilGen does require re-optimization to align with new objectives. We recognize this as a limitation, though it does not undermine our main contribution—demonstrating that utility-guided generation leads to highly effective synthetic data for the target task. We will dedicate our future work to making UtilGen more generalizable across tasks.
>
> > **Q6. Can this framework be extended to few-shot settings? (Weakness 5)**
>
> We appreciate this insightful question regarding the framework's adaptability to low-resource scenarios. As designed, **UtilGen naturally supports few-shot learning**, requiring only minimal labeled data while maintaining competitive performance. This capability is empirically validated in our comparative experiments with DataDream[4] on the ImageNet-100, where UtilGen demonstrates superior effectiveness in data-scarce regimes. Thank you for raising this important point, and we will include additional analysis of few-shot scenarios in our revised manuscript to further strengthen this discussion.
>
> **Experimental Setup**:
> - Half of the images are simultaneously used for:
>   - The training set for Weight-Net training
>   - Textual inversion optimization
> - The remaining half serves as the validation set for Weight-Net training.
>
>
> | Method       | 2-shot (%) | 4-shot (%) | 8-shot (%) | 16-shot (%) |
> |:-------|:--------:|:--------:|:-------:|:--------:|
> | DataDream[4] | 21.70      | 27.18      | 33.44      | 35.48       |
> | **UtilGen**  | **22.32**  | **29.16**  | **36.82**  | **39.62**   |
>
> **References**
>
> [1] Meta-Weight-Net: Learning an Explicit Mapping  For Sample Weighting
>
> [2] Diffusion Curriculum: Synthetic-to-Real Generative Curriculum Learning via Image-Guided Diffusion
>
> [3] Controlled Training Data Generation with Diffusion Models
>
> [4] DataDream: Few-shot Guided Dataset Generation
>
> Thank you once again for your valuable feedback, which has helped us to significantly improve the scientific rigor of our paper. We will incorporate the corresponding revisions and clarifications into the updated manuscript. If you have any further questions or concerns, we would be glad to address them.
>
> Sincerely,
>
> The Authors

---

### Note · Authors · 2025-08-11

Dear PC, SAC, AC, and Reviewers,

We sincerely thank all reviewers for their constructive feedback, which has greatly polished our manuscript. We are encouraged by the overall positive evaluation and thoughtful comments from all reviewers, and appreciate the valuable suggestions that helped us clarify and improve key aspects of our work. This broad agreement highlights the significance and robustness of our approach.

The core contribution of our work lies in **introducing a paradigm shift in data augmentation: rather than merely optimizing intrinsic data properties such as visual fidelity or diversity, we focus on optimizing the utility of synthetic data. This enables us to explicitly align the optimization objectives of data-generation models with downstream task requirements.** We clarify that our method can be directly applied in **few-shot scenarios**, requiring only a small amount of real labeled data for training the weight network and guiding data generation. At the same time, UtilGen demonstrates strong generalization and transferability across diverse classifier architectures and downstream tasks, highlighting its broad applicability.

In closing, we sincerely appreciate the invaluable suggestions from all reviewers, which have substantially improved the scientific rigor and clarity of our manuscript. We will carefully incorporate their comments into the final version if accepted, and believe our work offers a novel and effective paradigm for generative data augmentation.

Sincerely,
The Authors

---

### Decision · Program_Chairs · 2025-09-17

**Decision:**

Accept (poster)

**Comment:**

In this paper, the authors present a data augmentation technique that optimizes sample generation to maximize downstream task performance through a bilevel optimization problem.  The method is based on model-level optimization and instance-level optimization.  Their technique is extensively evaluated and shows consistent improvements over SOTA on a variety of tasks and architectures.  The paper has significant novelty and has the potential to lead to a shift in the philosophy of building data augmentation algorithms.